# Hybrid Autoencoders for Tabular Data: Leveraging Model-Based Augmentation in Low-Label Settings

**Erel Naor**
Bar-Ilan University
erel.naor@live.biu.ac.il

**Ofir Lindenbaum**
Bar-Ilan University
ofir.lindenbaum@biu.ac.il

## Abstract

Deep neural networks often underperform on tabular data due to sensitivity to irrelevant features and a spectral bias toward smooth, low-frequency functions, limiting their ability to capture sharp, high-frequency signals in low-label regimes. While self-supervised learning (SSL) holds promise in such settings, it remains challenging in tabular domains due to the limited availability of effective data augmentations. We introduce **TANDEM** (*Tree-And-Neural Dual Encoder Model*), a hybrid autoencoder that trains a neural encoder alongside an oblivious soft decision tree (OSDT) encoder, both guided by dedicated stochastic gating networks for sample-specific feature selection. The encoders share a decoder and are coupled via alignment losses, encouraging complementary yet consistent representations. The training-only use of the tree *operates as* model-based augmentation, nudging representations toward tabular-relevant features while preserving a lean inference path (only the neural encoder is deployed). Spectral analysis highlights distinct yet complementary inductive biases across encoders, and experiments on classification and regression benchmarks in low-label settings show consistent gains over strong deep, tree-based, and SSL baselines.

## 1 Introduction

In many real-world applications, tabular data is the dominant data format, especially in domains such as healthcare and finance. Tree-based methods, such as gradient-boosted decision trees (GBDT) [6], XGBoost [5], or CatBoost [14], are often the go-to models for classification tasks on tabular data, consistently delivering strong performance with minimal tuning. While deep neural networks have achieved impressive results in domains like vision and language, they often struggle to match the performance of tree-based models in tabular settings. One contributing factor is the spectral inductive bias of neural networks: they tend to favor smooth, low-frequency functions, which may not align well with the complex, irregular, and often high-frequency patterns found in tabular datasets [15]. This limitation becomes especially pronounced when modeling interactions between heterogeneous features, such as mixed categorical and numerical variables. Moreover, tabular datasets often include nuisance features—variables that are irrelevant or misleading in specific contexts. Neural networks often struggle to isolate and suppress these features, resulting in overfitting or poor generalization.

In many real-world tabular domains, such as healthcare, biology, and finance, labeled data is scarce and expensive, while unlabeled data is often abundant. This has led to growing interest in self-supervised learning (SSL) methods that can leverage unlabeled data to improve performance in low-label settings. However, applying SSL to tabular data is particularly challenging. Common augmentations such as noise injection or feature value swapping often distort critical feature relationships or create unrealistic samples, especially in the presence of categorical variables [18]. As a result, general-purpose augmentation strategies are challenging to design and often require dataset-specific tuning.

39th Conference on Neural Information Processing Systems (NeurIPS 2025).

To address this, Masked Autoencoders (MAEs) have been proposed as a more structure-preserving alternative. By learning to reconstruct selectively masked inputs, they provide a principled method for training models that does not rely on potentially unreliable augmentations. Yet even MAEs face limitations when applied to heterogeneous tabular data, where the semantics of categorical features may be lost or misrepresented during corruption and reconstruction [12].

To mitigate these challenges, we propose **TANDEM** (*Tree-And-Neural Dual Encoder Model*), an alternative strategy that shifts the focus from data augmentation to model enrichment. Our approach employs a hybrid self-supervised autoencoder architecture composed of two fundamentally different encoder types: a deep neural network (NN) and an oblivious soft decision tree (OSDT), both trained jointly through a shared decoder. Alignment losses are used to promote consistency across the representations learned by each encoder. The OSDT encoder introduces strong inductive biases that are particularly well suited to tabular data, capturing sharp, high-frequency patterns and conditional feature interactions. Through joint training, it shapes the NN encoder's representations, encouraging more structured and robust encoding. At inference time, only the NN encoder is used, preserving the flexibility and compatibility of neural networks with downstream tasks. By combining these complementary behaviors during training, the model learns a more effective latent space for few-shot classification without relying on predefined augmentation schemes.

In addition, we incorporate separate sample-specific stochastic gating networks for each encoder, which are trained jointly with the autoencoder using the same reconstruction objective. These gating networks function as dynamic, sample-dependent feature selectors, serving as model-specific data transformations. The gating mechanism selectively filters out less relevant variation while preserving the information necessary for accurate reconstruction, resulting in input transformations that preserve semantic structure while tailoring the feature space to the strengths of each encoder. Rather than distorting the data through fixed augmentations, the gating networks act as learnable, sample-specific filters that suppress irrelevant or noisy variation. This process helps the neural encoder overcome its bias toward smooth, low-frequency patterns, enabling both encoders to capture the signal more effectively in low-label regimes.

**Our main contributions are as follows:**

(1) We introduce **TANDEM**, a hybrid self-supervised autoencoder that combines a neural encoder, an oblivious soft decision tree encoder, a shared decoder, and sample-specific stochastic gating networks, enabling the learning of complementary representations suited to tabular data.
(2) We demonstrate that the representations learned by the neural encoder enable strong performance for both classification and regression tasks under low-label conditions, surpassing established deep learning and tree-based baselines.
(3) We conduct extensive experiments across a diverse suite of tabular datasets and systematically vary the number of labeled samples (from 50 to 1000 per dataset), establishing the robustness of TANDEM in a range of low-label regimes.
(4) We provide both qualitative spectral analysis and quantitative comparison of gating activations, revealing how the two encoders capture distinct and complementary inductive biases.

## 2   Related Work

**Regression and Classification on Tabular Data: Tree-Based and Deep Learning Models**   Tabular data remains central in real-world applications such as healthcare, finance, and recommendation systems. Classical approaches, including linear regression, logistic regression, and decision trees, have long been used due to their simplicity and interpretability. Among modern methods, ensemble-based models like GBDT, XGBoost, and CatBoost consistently dominate benchmarks, due to their ability to model nonlinear feature interactions and handle heterogeneous inputs with minimal tuning [6, 5, 14]. Motivated by the successes of deep learning in vision and language, several works have adapted neural architectures to tabular data. DeepTLF aligns numerical and categorical features [4], TabM simulates ensembles within compact MLPs [7], and TabPFN frames classification as probabilistic inference via synthetic pretraining [10]. SAINT and FT-Transformer apply attention to tabular structure [19, 8], while LSPIN introduces locally sparse neural networks with sample-specific masks [24]. Other lines of work incorporate tree-based structures into differentiable models. Soft Decision Trees (SDTs) [27] use smooth gating functions for gradient-based optimization.

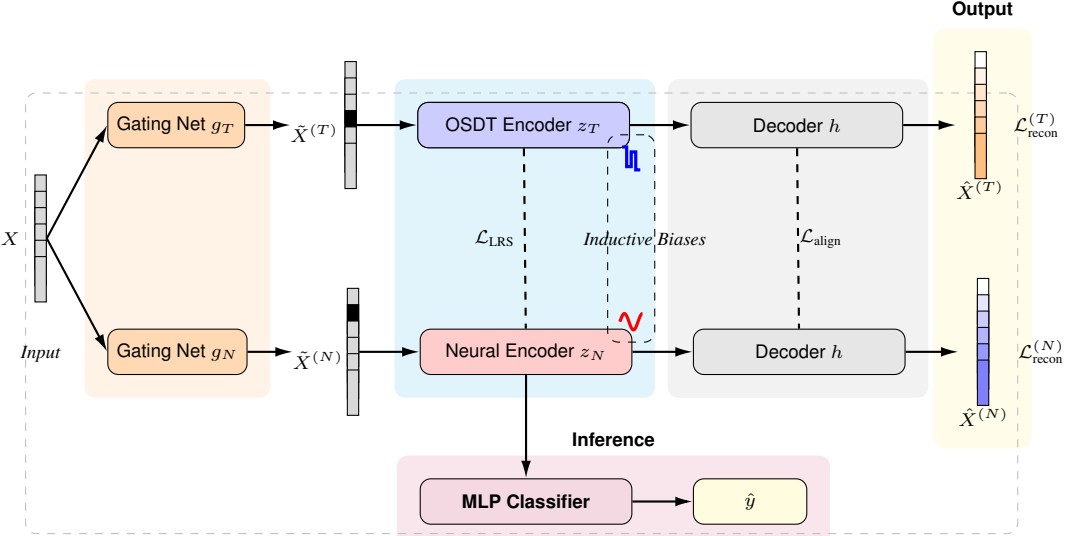

Figure 1: **Overview of the TANDEM architecture.** Input $X$ is augmented via two distinct stochastic gating networks, producing separate masked views for a neural encoder and an OSDT encoder, each of which is illustrated to the right to reflect their respective inductive biases. TANDEM is trained using reconstruction loss, alignment loss, and latent representation similarity (LRS) loss. During inference, only the neural encoder and the MLP classifier are used to predict the output label $\hat{y}$, based on the view gated by the neural encoder's gating net.

Oblivious Soft Decision Trees (OSDTs) [13] impose depth-wise consistency in splits, improving interpretability and stability. These models, however, rely solely on labeled data.

**Self-Supervised Learning in Tabular Data**  Several self-supervised methods apply reconstruction objectives to tabular data. SubTab reconstructs full inputs from masked subsets [21], VIME adds masking and noise with an auxiliary corruption-prediction task [25], and SCARF aligns clean and corrupted views via contrastive learning [2]. These methods rely on fixed corruption schemes or handcrafted views, which may not generalize across datasets. In contrast, our approach applies learnable, sample-specific transformations tailored to the input distribution. Tree-based autoencoders have also been proposed, using differentiable trees as both encoder and decoder [26]. While these models reduce dimensionality and capture hierarchy, they are constrained by tree inductive biases. Unlike ours, they lack the flexibility and generalization of neural networks. TANDEM addresses this by combining tree-based and neural encoders to capture complementary inductive biases.

**Unsupervised Feature Selection for Tabular Data**  Unsupervised feature selection identifies informative features without requiring labels, often for tasks such as reconstruction or clustering. Regularized autoencoders learn global feature subsets for reconstruction [17]. More recent approaches use stochastic gating networks [20], which learn per-sample, input-dependent soft masks for context-aware selection. While prior work uses gating for sparsity or interpretability, we reinterpret them as a learnable augmentation mechanism. In TANDEM, gating networks are trained jointly with cross-reconstruction: one is applied globally to the neural encoder input, and others are hierarchically applied across OSDT layers. This enables model-specific and sample-specific views that suppress nuisance features and emphasize the shared, informative structure for reconstruction and downstream learning.

## 3  Method

We address semi- and self-supervised representation learning for tabular data with scarce labels. Let the unlabeled pool be $D_{\text{unlab}} = \{x_n\}_{n=1}^{N} \subset \mathbb{R}^D$ and the labeled set $D_{\text{lab}} = \{(x_m, y_m)\}_{m=1}^{M}$, where $M \ll N$. Our goal is to learn useful embeddings from $D_{\text{unlab}}$ that improve downstream classification or regression on $D_{\text{lab}}$.

**Model.** **TANDEM** is a hybrid masking autoencoder with *complementary encoders* and a *shared decoder*. Each input $x \in \mathbb{R}^D$ is first passed through a sample-specific gating network (STG) that outputs a feature mask $g(x) \in [0,1]^D$. The masked view $\tilde{x} = x \odot g(x)$ is fed in parallel to: (i) a fully connected neural encoder producing $z^{\mathrm{NN}} \in \mathbb{R}^k$, and (ii) an ensemble of Oblivious Soft Decision Trees (OSDTs) producing $z^{\mathrm{OSDT}} \in \mathbb{R}^k$. A shared decoder $h$ reconstructs to $\hat{x}^{\mathrm{NN}} = h(z^{\mathrm{NN}})$ and $\hat{x}^{\mathrm{OSDT}} = h(z^{\mathrm{OSDT}})$, both in $\mathbb{R}^D$. Over a batch, this yields $Z^{\mathrm{NN}}, Z^{\mathrm{OSDT}} \in \mathbb{R}^{B \times k}$ and $\hat{X}^{\mathrm{NN}}, \hat{X}^{\mathrm{OSDT}} \in \mathbb{R}^{B \times D}$.

**Training and use.** Pretraining is performed on $D_{\mathrm{unlab}}$ using the masked-reconstruction objective (with auxiliary consistency terms defined later). After pretraining, downstream heads are trained on $D_{\mathrm{lab}}$. At inference, only the neural encoder and a lightweight predictor are used. The training objective combines three components: a reconstruction loss for each autoencoder, an alignment loss between their reconstructions, and a similarity loss between the latent representations. Together, these losses regularize the model and promote consistency across encoders with different inductive biases. This design enables the neural encoder to leverage the structured, high-frequency inductive bias of the OSDT encoder, as further demonstrated in our spectral analysis 5.

## 3.1 Oblivious Soft Decision Tree Encoder

The OSDT encoder consists of an ensemble of shallow, differentiable binary decision trees [27] with fixed depth $L$. Each tree follows the *oblivious* decision tree structure, where all internal nodes at the same level share a single projection vector. This constraint ensures a consistent, hierarchical partitioning of the input space, thereby reducing variability in decision logic across paths. At each tree level $\ell \in \{1, \ldots, L\}$, a learned projection vector $w_\ell \in \mathbb{R}^D$ is used to compute a soft split score from the (potentially gated) input $x \in \mathbb{R}^D$, using the equation $s_\ell(x) = \langle w_\ell, x \rangle - \tau_\ell$, where $\tau_\ell \in \mathbb{R}$ is a learned threshold. The score is passed through a temperature-scaled sigmoid to produce left/right routing probabilities: $\sigma_\ell^\pm(x) = \sigma\left(\pm \frac{s_\ell(x)}{T_\ell}\right)$. Each of the $2^L$ leaf nodes corresponds to a binary code $b \in \{0,1\}^L$, and the probability of reaching a given leaf is computed as:

$$p_{\mathrm{leaf}}(x) = \prod_{\ell=1}^{L} \left[\sigma_\ell^+(x)\right]^{b_\ell} \cdot \left[\sigma_\ell^-(x)\right]^{1-b_\ell}.$$

Each tree produces a soft distribution over its $2^L$ leaves, representing the probability of reaching each path. These leaf probabilities are concatenated to form the tree's latent output, denoted by $f_t^{\mathrm{OSDT}}(x) \in \mathbb{R}^{2^L}$. The final representation of the encoder is obtained by averaging these vectors across all $T$ trees:

$$z^{\mathrm{OSDT}}(x) = \frac{1}{T} \sum_{t=1}^{T} f_t^{\mathrm{OSDT}}(x) \in \mathbb{R}^{2^L}.$$

This vector $z^{\mathrm{OSDT}}$ serves as the latent representation of the tree encoder and is passed to the shared decoder. For further details on oblivious differentiable tree-based encoders, we refer readers to the NODE paper [13].

## 3.2 Training Objective

TANDEM's training objective is designed to align the two encoder streams while preserving their individual inductive biases. It combines three loss components: a reconstruction loss for each encoder, an alignment loss between the reconstructions, and a term measuring the similarity of the latent representations. The reconstruction loss encourages each encoder to preserve semantic structure from its gated input, ensuring that both the neural and tree-based views remain informative:

$$\mathcal{L}_{\mathrm{recon}} = \frac{1}{N} \sum_{m=1}^{N} \left( \|x_m - \hat{x}_m^{\mathrm{OSDT}}\|_2^2 + \|x_m - \hat{x}_m^{\mathrm{NN}}\|_2^2 \right),$$

where $x_m$ is the input vector of the $m$-th sample, and $\hat{x}_m^{\mathrm{NN}}$, $\hat{x}_m^{\mathrm{OSDT}}$ are the reconstructed outputs of the neural and OSDT encoders, respectively. To promote consistency between the outputs of the two

encoders, we apply an alignment loss between their reconstructions:

$$\mathcal{L}_{\text{align}} = \frac{1}{N} \sum_{m=1}^{N} \|\hat{x}_m^{\text{OSDT}} - \hat{x}_m^{\text{NN}}\|_2^2.$$

Finally, we enforce agreement in the latent space by minimizing the average cosine distance between the latent vectors produced by each encoder:

$$\mathcal{L}_{\text{LRS}} = \frac{1}{N} \sum_{m=1}^{N} \left( 1 - \frac{\langle z_m^{\text{NN}}, z_m^{\text{OSDT}} \rangle}{\|z_m^{\text{NN}}\|_2 \cdot \|z_m^{\text{OSDT}}\|_2} \right),$$

where $z_m^{\text{NN}}$ and $z_m^{\text{OSDT}}$ are the latent vectors of the $m$-th sample, produced by the neural and OSDT encoders, respectively. The combined objective balances encoder-specific reconstruction with cross-view consistency, encouraging complementary yet compatible representations.

### 3.3 Stochastic Gating Network as Sample-Level Masking

We implement a sample-specific gating mechanism that selects input features through stochastic, differentiable masks. Given an input $x \in \mathbb{R}^D$, a neural gating network $f_\theta(x)$ produces a parameter vector $\mu(x) \in \mathbb{R}^D$. A gating vector $g(x) \in [0,1]^D$ is then sampled using a clipped Gaussian perturbation:

$$g(x) = \max(0, \min(1, 0.5 + \mu(x) + \epsilon)), \quad \epsilon \sim \mathcal{N}(0, \sigma^2).$$

We fix $\sigma = 0.5$ throughout training, as suggested in [23, 11]. The injected noise encourages the values of $g_d(x)$ toward binary decisions, while preserving gradient flow.

$$\tilde{x} = x \odot g(x).$$

In the neural encoder, a single gating network $f_{\theta^{\text{NN}}}$ computes a global mask. In contrast, the OSDT encoder uses a distinct gating network at each tree level $\ell \in \{1, \ldots, L\}$, producing a level-specific mask $g_\ell^{\text{OSDT}}(x) \in [0,1]^D$. This mask is applied before computing the soft split score:

$$\tilde{x}_\ell = x \odot g_\ell^{\text{OSDT}}(x), \quad s_\ell(x) = \langle w_\ell, \tilde{x}_\ell \rangle - \tau_\ell.$$

This structure allows the OSDT encoder to learn different feature selections at different depths, supporting hierarchical and progressively refined decisions. Although the gating networks are parameterized and trained independently within each encoder, they are jointly guided via the shared decoder and alignment objectives. This coordination implicitly aligns the gating behavior across encoders, while allowing each to exploit its architectural inductive bias.

## 4 Experiments and Results

We evaluate whether combining neural and tree-based encoders with a sample-specific gating mechanism improves performance on both classification and regression tasks in tabular datasets, particularly when labeled examples are scarce. We aim to assess the utility of self-supervised pretraining for generating effective representations under such low-supervision settings.

### 4.1 Dataset Selection and Preprocessing

Our experiments are conducted in the low-label regime where labeled budgets range from 50 to 1000 samples per dataset, enabling consistent evaluation under limited supervision. We used all relevant classification datasets from OpenML analyzed in two widely cited studies on the limitations of deep learning for tabular data [9, 18]. We retained only datasets with at least 2,500 samples *per class*. For each class, 2,000 samples were allocated for self-supervised pretraining, and up to 1,000 labeled samples in total (across classes) were reserved for downstream evaluation. This filtering resulted in 19 classification datasets. For regression, we extracted 13 datasets that satisfy the same filtering as in the classification benchmark from the same OpenML sources referenced above. All datasets were preprocessed using a fixed pipeline: categorical features were one-hot encoded, numerical features were min-max normalized to $[0, 1]$, and missing values were imputed with the column-wise mean.

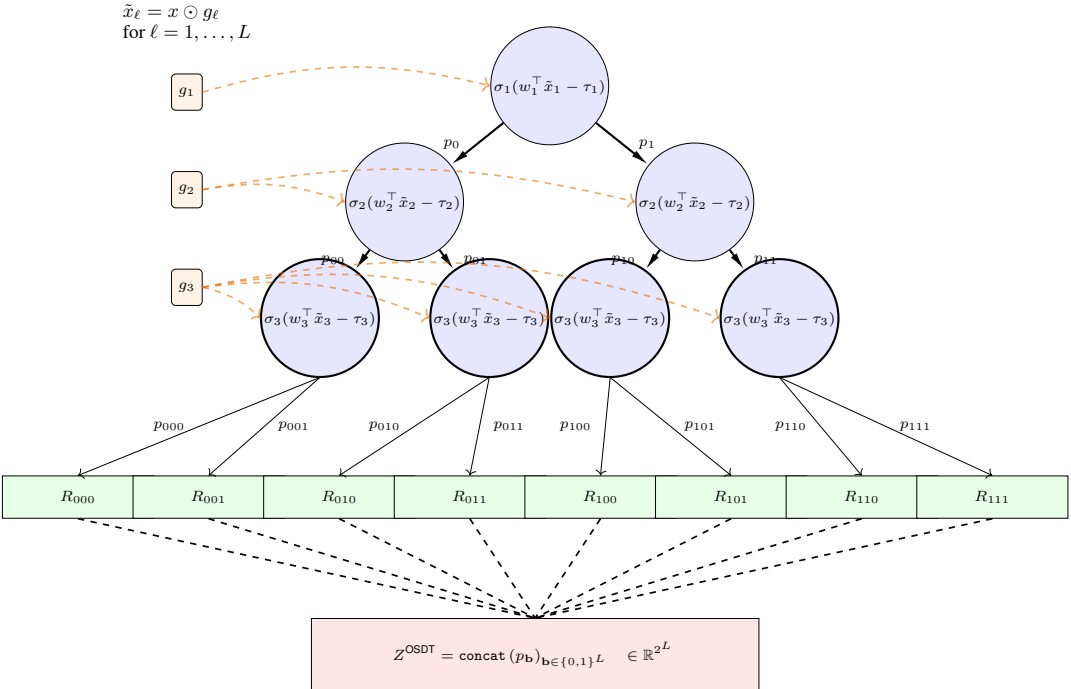

Figure 2: Architecture of the OSDT encoder in TANDEM. At each depth $\ell \in \{1, \ldots, L\}$, the input $x$ is gated by a dedicated gating network $g_\ell$, producing a masked vector $\tilde{x}_\ell = x \odot g_\ell$. This masked input is projected by a learnable vector $w_\ell$ and compared against a threshold $\tau_\ell$, producing a soft binary decision. Probabilities propagate through the tree to define $p_{\mathbf{b}}$, the soft path probability to leaf $R_{\mathbf{b}}$. The final output of the encoder is the concatenation of all leaf probabilities, $Z^{\text{OSDT}} \in \mathbb{R}^{2^L}$, serving as a disentangled latent representation.

## 4.2 Training and Evaluation Setup

Pretraining was run for 100 epochs with a batch size of 128, and this configuration was held constant across all experiments to ensure fair and consistent comparison. We used RMSprop as the optimizer for both pretraining and fine-tuning across all models. The training objective combined reconstruction, output alignment, and latent similarity losses, as described in Section 3. Hyperparameters, including learning rate, encoder depth, and weight decay, were selected using Optuna over 50 trials based on the validation loss. Complete optimization, including computational details and parameter ranges, is provided in the appendix.

For downstream evaluation, a single-layer MLP classifier or regressor was trained on the labeled subset using the neural encoder. The encoder was frozen for the first 25 epochs, then fine-tuned for an additional 25 epochs at a reduced learning rate. Early stopping was based on validation accuracy or MSE, as appropriate. If present, the gating network was kept frozen during fine-tuning and used as a per-sample feature selector; it consisted of a two-layer MLP with `tanh` activations and a hard-sigmoid output that produced binary masks.

We evaluate two groups of baselines under identical data budgets. *Supervised-only* methods are trained solely on the labeled subset and include multinomial logistic regression (classification), linear regression (regression), a standard MLP, XGBoost [5], CatBoost [14], TabM [7], DeepTLF [4], and TabNet (supervised) [1]. *Self-supervised + fine-tuning* methods are pretrained on the same unlabeled pool we use (2,000 samples per class) and then fine-tuned on the labeled subset; this group includes VIME [25], SCARF [2], SubTab [21], and TabNet with self-supervised pretraining [1]. Following the authors' guidance, *TabPFN* [10] is used only for classification. For *TabNet*, we report the stronger of its supervised and self-supervised variants for each task. Across all methods, hyperparameters are tuned using Optuna (50 trials), and model selection is based on validation performance (accuracy for classification and MSE for regression). For component analysis, we report ablations of our approach: (1) SS-AE, (2) SS-AE + gating, (3) SS-AE with an OSDT encoder and

Table 1: Comparison across models on **classification** datasets (400 labeled samples). Best results per row are in **bold**.

| Dataset | MLogReg | TabNet | DeepTLF | TabM | TabPFN | XGBoost | CatBoost | MLP | TANDEM |
|---|---|---|---|---|---|---|---|---|---|
| CP | 0.4927 | 0.6000 | 0.5438 | 0.5827 | 0.5998 | 0.5822 | 0.5401 | 0.5792 | **0.6779** |
| MT | 0.7521 | 0.6781 | 0.6757 | 0.7720 | 0.8139 | 0.7868 | 0.7929 | 0.7798 | **0.8180** |
| OG | 0.6228 | 0.5736 | 0.3712 | 0.6228 | 0.6514 | 0.6136 | 0.6363 | 0.6321 | **0.6870** |
| PW | 0.9373 | 0.5906 | 0.9281 | 0.9384 | 0.9477 | 0.9353 | 0.9327 | 0.9315 | **0.9618** |
| AD | 0.7992 | 0.6583 | 0.7645 | 0.8115 | 0.8199 | 0.7875 | 0.8019 | 0.8006 | **0.8200** |
| ALB | 0.6065 | 0.5708 | 0.5512 | 0.6196 | 0.6494 | 0.6096 | 0.6354 | 0.5929 | **0.7038** |
| BM | **0.8325** | 0.5458 | 0.6306 | 0.8132 | 0.8241 | 0.7991 | 0.8171 | 0.7913 | 0.8233 |
| CO | 0.4624 | 0.4618 | 0.4881 | 0.5049 | 0.5485 | 0.5326 | 0.4975 | 0.4963 | **0.5491** |
| CC | 0.6169 | 0.5458 | 0.6302 | 0.6218 | 0.6451 | 0.6780 | 0.6690 | 0.7190 | **0.7331** |
| EL | 0.6617 | 0.5391 | 0.6424 | 0.6610 | **0.7723** | 0.7668 | 0.7654 | 0.7525 | 0.6940 |
| HE | 0.4593 | 0.3779 | 0.3844 | 0.4811 | —* | 0.4590 | 0.4853 | 0.4448 | **0.5462** |
| HI | 0.5454 | 0.4984 | 0.4965 | 0.5532 | **0.6499** | 0.6096 | 0.6069 | 0.6035 | 0.6459 |
| JA | 0.5105 | 0.4930 | 0.4649 | 0.5743 | **0.5986** | 0.5409 | 0.5413 | 0.5417 | 0.5660 |
| NU | 0.4833 | 0.5016 | 0.4932 | 0.5221 | 0.4333 | 0.5308 | 0.5231 | 0.5517 | **0.6545** |
| RS | 0.6992 | 0.5063 | 0.6590 | **0.7886** | 0.7554 | 0.7057 | 0.7328 | 0.7243 | 0.7576 |
| VO | 0.4624 | 0.4389 | 0.3966 | 0.4790 | 0.5082 | 0.4736 | 0.4975 | **0.5400** | 0.5220 |
| POL | 0.8515 | 0.7511 | 0.5000 | 0.7480 | 0.9520 | 0.9425 | 0.9330 | 0.9321 | **0.9538** |
| CA | 0.8400 | 0.6667 | 0.6930 | 0.8107 | 0.8703 | 0.8416 | 0.8400 | 0.8330 | **0.8921** |
| EY | 0.4862 | 0.3655 | 0.3753 | 0.5058 | 0.5811 | 0.5461 | 0.5400 | 0.5245 | **0.5928** |
| **Mean Accuracy** | 0.6380 | 0.5454 | 0.5626 | 0.6532 | 0.7012 | 0.6706 | 0.6731 | 0.6721 | **0.7124** |
| **Mean Rank** | 6.16 | 8.00 | 8.05 | 4.84 | 2.56 | 4.47 | 4.16 | 4.84 | **1.58** |

*TabPFN supports a maximum of 10 classes and could not be evaluated on the *helena* dataset (13 classes); TabPFN's mean/rank exclude that dataset.

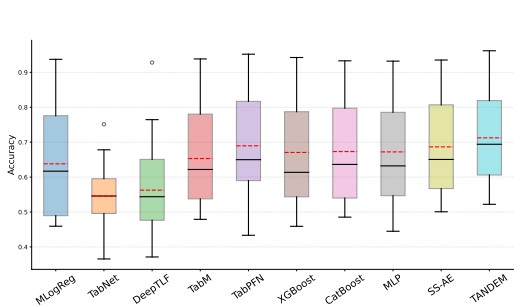

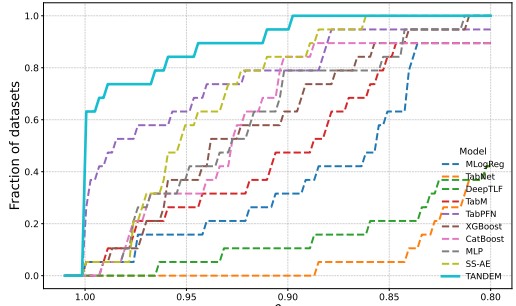

Figure 3: **Classification accuracy distribution across models.** Boxplot across baseline models and **TANDEM**; red lines denote the mean and black lines denote the median.

Figure 4: **Classification Dolan–Moré profiles.** Model accuracy relative to the per-dataset best; higher curves indicate stronger performance across datasets.

neural decoder, (4) TANDEM without gating, and (5) TANDEM without the latent similarity and alignment losses; all ablations share the same decoder and architectural settings.

### 4.3 Experimental Setup

Each experiment was repeated 100 times with varying seeds and splits. We report mean accuracy (classification) or mean MSE (regression) per dataset, along with aggregated mean accuracy, mean MSE, and mean rank across all datasets. Additional statistics, including standard deviations and further details on datasets, are provided in the appendix.

### 4.4 Benchmarks (Classification & Regression)

**Performance against baselines at 400 labels.** Table 1 and Figure 3 (classification, 400 labels) show that TANDEM achieves the highest mean accuracy and the best average rank, with TabPFN as the closest competitor. For regression at the same label budget, Table 2 and Figure 5 indicate that TANDEM attains the lowest mean MSE and the best mean rank, with XGBoost as the nearest competitor. **In both tasks,** TANDEM is competitive or best on most datasets and delivers the strongest overall results; the Dolan–Moré curves [16] (Figures 4, 6) further underscore its robustness across datasets.

Table 2: Comparison across models on **regression** datasets (400 labeled samples). Best results per row are in **bold**.

| Dataset | CatBoost | LogReg | MLP | SCARF | SubTab | VIME | TabM | TabNet | XGBoost | TANDEM |
|---|---|---|---|---|---|---|---|---|---|---|
| BSD | 0.1506 | 0.6088 | 0.5395 | 0.7377 | 0.5778 | 0.5699 | 0.4922 | 0.7634 | 0.1742 | **0.0722** |
| BH | 0.0291 | 0.0020 | 0.0382 | 0.0382 | 0.0047 | 0.0347 | 0.0012 | 0.0703 | 0.0302 | **0.0011** |
| MBGM | 0.5206 | 2.0242 | 0.6508 | 0.7517 | 0.6400 | 0.7208 | 0.6666 | 1.0083 | 0.5612 | **0.3354** |
| SGEM | 0.0340 | 0.0041 | **0.0017** | 0.0038 | 0.0113 | 0.1848 | 0.0032 | 0.0231 | 0.0171 | 0.0025 |
| AS | 0.0329 | 0.5474 | 0.0515 | 0.1331 | 0.3336 | 0.7107 | 0.0487 | 0.2463 | **0.0264** | 0.0297 |
| BF | **0.6565** | 0.8731 | 0.8931 | 0.8461 | 0.8874 | 0.9215 | 0.8409 | 0.9067 | 0.6986 | 1.0057 |
| CO | **0.7047** | 0.8370 | 0.7711 | 0.8030 | 0.7869 | 0.8152 | 0.8055 | 0.9399 | 0.7395 | 0.7334 |
| DI | 0.0889 | 0.2330 | 0.0531 | 0.1252 | 0.1102 | 0.1354 | **0.0475** | 0.2908 | 0.0675 | 0.0498 |
| EL | 0.1645 | 0.5115 | 0.2290 | 0.2244 | 0.2331 | 0.1732 | **0.1549** | 0.2013 | **0.1549** | 0.1835 |
| EY | 0.4987 | 0.5237 | 0.5533 | 0.6110 | 0.5825 | 0.5591 | 0.5577 | 0.6253 | 0.4805 | **0.4143** |
| HS | 0.3761 | 0.3960 | 0.2664 | 0.3198 | 0.3519 | 0.4175 | 0.5212 | 0.3923 | 0.2950 | **0.1436** |
| VS | 0.0015 | 0.1682 | **0.0013** | 0.0094 | 0.0086 | 0.3711 | 0.0030 | 0.0483 | 0.0068 | 0.0024 |
| YP | 1.0557 | 2.2639 | **1.0278** | 1.0804 | 1.1325 | 1.1236 | 1.0419 | 1.2104 | 1.1751 | 1.2300 |
| **Mean MSE** | 0.3318 | 0.6918 | 0.3877 | 0.4372 | 0.4354 | 0.5183 | 0.4006 | 0.5174 | 0.3405 | **0.3234** |
| **Mean Rank** | 4.00 | 8.38 | 4.38 | 7.23 | 7.08 | 8.46 | 4.85 | 9.38 | 4.15 | **3.38** |

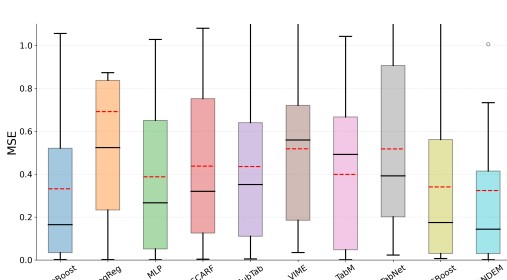

Figure 5: **Regression MSE distribution across models.** Boxplot across baseline models and **TANDEM**; red lines denote the mean and black lines denote the median.

Figure 6: **Regression Dolan–Moré profiles.** Model MSE relative to the per-dataset best; higher curves indicate stronger performance across datasets.

**Ablation studies (shared design).** Tables 3 (classification) and 4 (regression) assess matched ablations under the 400-label setting. Across both tasks, removing gating or omitting either encoder degrades performance, whereas full TANDEM, combining the neural and tree encoders with gating, consistently yields the strongest results.

**Performance across label budgets (50–1000 labels).** Figure 7 tracks mean performance as label budgets increase from 50 to 1000 in both classification (left) and regression (right). Across both tasks, TANDEM is consistently either the top method or competitive with the best throughout the low-label regime (50–1000 labels).

# 5 Understanding Complementary Gating Behaviors via Frequency Decomposition

To better understand the role of sample-specific gating networks, we analyze how gating affects the spectral composition of the input within a single class. For each dataset, we identify the class in which TANDEM achieves the highest classification accuracy and restrict our analysis to samples from that class. We then compute the unnormalized discrete Fourier transform (NUDFT) over the 50 most variant features, comparing the resulting frequency distributions across different gated and ungated versions of the input.

To analyze the effect of gating on the spectral composition of the input, we compare the spectrum of the original input $x$ to its transformed versions under different gating mechanisms. Specifically, we evaluate the gated input $\tilde{x}^{\mathrm{NN}} = x \odot g^{\mathrm{NN}}(x)$ produced by the stochastic gating network connected to the neural encoder (as defined in Section 3.3), and the input gated by the tree-based mechanism

Table 3: Ablation study comparing TANDEM and its variants on classification datasets.

| Dataset | SS-AE | SS-AE + Gating | OSDT AE + Gating | TANDEM (no gate) | TANDEM (no LRS + Alignment) | TANDEM |
|---|---|---|---|---|---|---|
| CP | 0.6505 | 0.6602 | 0.6321 | 0.6740 | **0.6866** | 0.6779 |
| MT | 0.7996 | 0.8096 | 0.7639 | 0.8117 | 0.8173 | **0.8180** |
| OG | 0.6500 | 0.6740 | 0.6601 | 0.6659 | 0.6815 | **0.6870** |
| PW | 0.9353 | 0.9353 | 0.8259 | 0.9464 | 0.8673 | **0.9618** |
| AD | **0.8202** | 0.8085 | 0.7688 | 0.8111 | 0.8035 | 0.8200 |
| ALB | 0.6497 | 0.6797 | 0.6652 | 0.6763 | 0.6993 | **0.7038** |
| BM | 0.8141 | 0.8141 | 0.7552 | 0.8138 | 0.8198 | **0.8233** |
| CO | 0.5007 | 0.5207 | 0.5373 | 0.5220 | 0.5475 | **0.5491** |
| CC | 0.6833 | 0.6900 | 0.6892 | **0.7434** | 0.7394 | 0.7331 |
| EL | 0.7429 | 0.7429 | 0.6149 | 0.7134 | 0.7078 | 0.6940 |
| HE | 0.5250 | 0.5350 | 0.4416 | 0.5315 | 0.5349 | **0.5462** |
| HI | 0.6179 | 0.6279 | **0.6621** | 0.6300 | 0.6529 | 0.6459 |
| JA | 0.5168 | 0.5468 | 0.5012 | 0.5529 | 0.5365 | **0.5660** |
| NU | 0.6092 | 0.6092 | 0.6140 | 0.6325 | 0.6067 | **0.6545** |
| RS | 0.7016 | 0.7016 | 0.7252 | 0.7311 | 0.7200 | **0.7576** |
| VO | 0.5154 | **0.5254** | 0.4334 | 0.4741 | 0.4840 | 0.5220 |
| POL | 0.9325 | 0.9381 | 0.9102 | 0.9212 | 0.9420 | **0.9538** |
| CA | 0.8653 | 0.8393 | 0.8493 | 0.8721 | 0.8803 | **0.8921** |
| EY | 0.5148 | 0.5296 | 0.4901 | 0.5124 | 0.5180 | **0.5928** |
| **Mean Accuracy** | 0.6815 | 0.6941 | 0.6600 | 0.6966 | 0.6971 | **0.7124** |
| **Mean Rank** | 4.45 | 3.61 | 4.71 | 2.92 | 2.79 | **1.74** |

Table 4: Ablation study comparing TANDEM and its variants on regression datasets.

| Dataset | SS-AE | SS-AE + Gating | OSDT AE + Gating | TANDEM (no gate) | TANDEM (no LRS + Alignment) | TANDEM |
|---|---|---|---|---|---|---|
| BSD | 0.2028 | 0.8266 | 0.9660 | 0.7938 | 0.7824 | **0.0722** |
| BH | 0.0026 | 0.0331 | 0.1088 | 0.0462 | 0.0367 | **0.0011** |
| MBGM | **0.3325** | 0.7409 | 0.9492 | 0.7630 | 0.7369 | 0.3354 |
| SGEM | 0.0044 | 0.3291 | 0.9896 | 0.1388 | 0.1756 | **0.0025** |
| AS | **0.0270** | 0.9236 | 0.9438 | 0.4520 | 0.7320 | 0.0297 |
| BF | 1.0890 | 0.9628 | 0.9966 | **0.9010** | 0.9286 | 1.0057 |
| CO | 0.7849 | 0.8479 | 1.0007 | 0.8828 | 0.9214 | **0.7334** |
| DI | 0.0626 | 0.3317 | 0.9764 | 0.3124 | 0.2010 | **0.0498** |
| EL | 0.1865 | 0.2380 | 0.4752 | 0.2651 | 0.2971 | **0.1835** |
| EY | 0.4944 | 0.7872 | 1.4029 | 0.6590 | 0.6402 | **0.4143** |
| HS | **0.1364** | 0.5933 | 1.0476 | 0.6383 | 0.7061 | 0.1436 |
| VS | 0.0023 | 0.6884 | 0.9759 | 0.5521 | 0.1157 | **0.0024** |
| YP | 1.3912 | 1.2134 | **1.1151** | 1.6014 | 1.3083 | 1.2300 |
| **Mean MSE** | 0.3628 | 0.6551 | 0.9207 | 0.5397 | 0.5832 | **0.3235** |
| **Mean Rank** | 2.69 | 4.15 | 5.50 | 4.00 | 3.73 | **1.92** |

in the OSDT encoder. Since the OSDT encoder employs a distinct gating network $g_{\ell,t}^{\text{OSDT}}(x)$ at each depth level $\ell \in \{1, \ldots, L\}$ for each tree $t \in \{1, \ldots, T\}$, we compute the aggregated gating signal by averaging all gating masks across all levels and trees:

$$\bar{g}^{\text{OSDT}}(x) = \frac{1}{T \cdot L} \sum_{t=1}^{T} \sum_{\ell=1}^{L} g_{\ell,t}^{\text{OSDT}}(x),$$

and apply it elementwise to obtain $\tilde{x}^{\text{OSDT}} = x \odot \bar{g}^{\text{OSDT}}(x)$. In addition, we analyze the spectrum of $\tilde{x}^{\text{NN}}$ produced by a SS-AE with the same neural gating network as used in TANDEM. All spectral distributions are shown on a shared axis for comparison against the original input.

Across datasets, all gating mechanisms function as adaptive frequency filters, suppressing high-frequency variation to varying degrees. We find that the neural gating transformation $\tilde{x}^{\text{NN}} = x \odot g^{\text{NN}}(x)$ in TANDEM consistently reduces high-frequency components more strongly than the tree-based transformation $\tilde{x}^{\text{OSDT}} = x \odot \bar{g}^{\text{OSDT}}(x)$. This observation aligns with prior work on the spectral bias of neural networks toward low-frequency representations [15], which can be limiting in tabular settings that often rely on sharper, high-frequency decision boundaries [3].

Notably, the neural and tree-based gating transformations within TANDEM, $\tilde{x}^{\text{NN}}$ and $\tilde{x}^{\text{OSDT}}$, exhibit the strongest spectral contrast. The neural gating $g^{\text{NN}}$ acts as a strong smoother, while the aggregated tree-based gating $\bar{g}^{\text{OSDT}}$ retains more high-frequency content. This reflects their complementary inductive biases, with each capturing different aspects of the input structure important for tabular decision boundaries. It is this consistent difference between $g^{\text{NN}}$ and $\bar{g}^{\text{OSDT}}$ in TANDEM that contributes to the formation of complementary representations in the shared latent space.

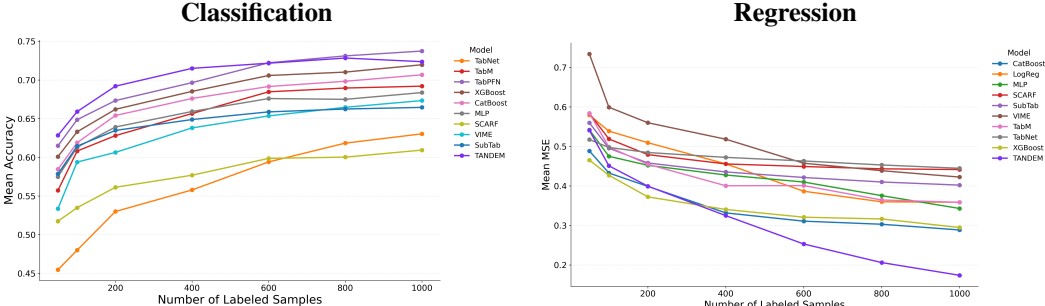

Figure 7: **Learning curves across label budgets.** Mean accuracy (left, classification) and mean MSE (right, regression) vs. label budget.

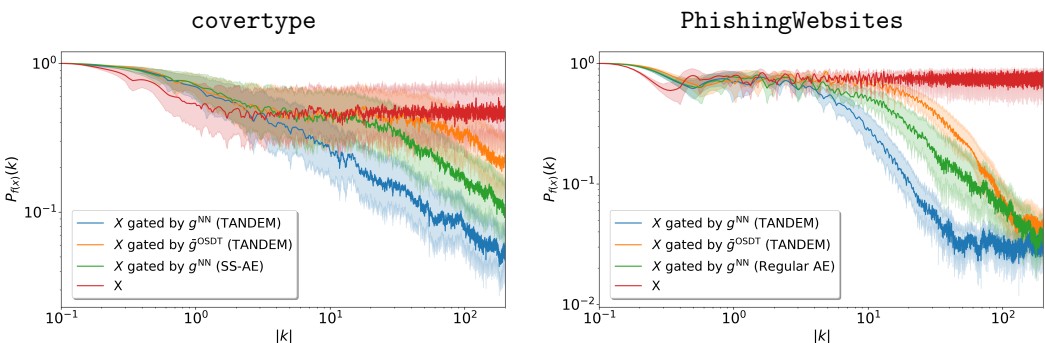

Figure 8: **Frequency spectra of gated inputs.** Comparison across datasets for each encoder-gated input using NUDFT.

# 6 Conclusion

We introduced **TANDEM**, a hybrid self-supervised framework designed for tabular data. This framework combines neural and tree-based encoders with sample-specific gating. TANDEM demonstrates impressive performance across various benchmarks in low-label scenarios, outperforming both neural and tree-based baselines.

Ablation and spectral analyses indicate that the two encoders offer complementary inductive biases, favoring smooth patterns and high-frequency patterns, respectively. This design enables TANDEM to generalize effectively from a limited amount of labeled data. The gating networks further enhance this by creating tailored, sample-specific input views that leverage the strengths of each encoder.

While our study has limitations, such as small sample sizes and a focus on low-label settings, these constraints reflect real-world challenges in tabular data and facilitate meaningful comparisons with specialized methods, such as TabPFN, which is also aimed at few-shot classification. Despite these limitations, our results demonstrate that small-scale evaluations can offer valuable insights into model behavior and inform the development of scalable solutions.

For future work, we believe that the architectural principles behind TANDEM of hybrid encoders, sample-specific gating, and model-based augmentation can be incorporated into transformer-based tabular models. This integration could help create more robust and interpretable tabular foundation models, in line with the vision presented in the recent position paper [22].

## Acknowledgment

OL was supported by the MOST grant No. 0007341.

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

## Checklist

1. Did you state the main claims of the paper in the abstract and introduction? [Yes]

2. Did you discuss the limitations of your work? [Yes] We address this point in the conclusion section.

3. Did you state the full set of assumptions of all theoretical results? [NA] The paper does not contain theoretical results.

4. Did you include complete proofs of all theoretical results? [NA]

5. If you ran experiments, did you include the code, data, and instructions needed to reproduce the main experimental results? [Yes] We include code to extract datasets and reproduce results as part of the supplementary material.

6. Did you include the full details of your experimental setup, including hyperparameters and training procedures? [Yes] Training setup is in Experiments and Results and hyperparameters are in Appendix A

7. Did you report error bars and describe the variability of your results? [Yes] Error bars (standard deviation over multiple runs) are reported in all performance tables. See Appendix B.

8. Did you provide information about compute resources used? [Yes] Details are provided in the appendix C.

9. Did you read the NeurIPS Code of Ethics and ensure your paper conforms to it? [Yes]

10. Did you discuss potential negative societal impacts of your work? [NA] Our work is purely methodological and does not involve real-world applications at this stage.

11. Did you describe safeguards for responsible release of high-risk models? [NA] Our models are not considered high-risk or dual-use.

12. If you used existing assets, did you respect their licenses and cite them appropriately? [Yes] All datasets and libraries are properly cited, including license details in Appendix B.

13. Did you document any new assets (e.g., code or datasets) and include details on licensing and consent? [Yes] Code and dataset handling scripts are included as part of the supplementary material

14. If you used crowdsourcing or conducted research with human subjects, did you include participant instructions and compensation details? [NA] No human participants or crowd-sourced data were used.

15. Did you obtain IRB approval (if required)? [NA] No human subjects were involved.

16. If you used LLMs in a non-trivial way, did you disclose their usage? [NA] No large language models were used in non-trivial manner.

# Appendix

# A    Results (400 labeled samples)

This section summarizes performance at a fixed budget of 400 labeled samples: baseline classification results (Table A.1), classification ablations (Table A.2), and classification significance counts (Table A.3); followed by baseline regression results (Table A.4), regression ablations (Table A.5), and regression significance counts (Table A.6).

## A.1    Classification

Table A.1: Baseline classification accuracy at 400 labeled samples (mean ± std).

| Dataset | LogReg | DeepTLF | TabM | TabPFN | XGBoost | CatBoost | MLP | VIME | SCARF | SubTab | TANDEM |
|---|---|---|---|---|---|---|---|---|---|---|---|
| CP | 0.4927 ± 0.0138 | 0.5438 ± 0.0390 | 0.5827 ± 0.0283 | 0.5998 ± 0.0294 | 0.5822 ± 0.0298 | 0.5401 ± 0.0114 | 0.5792 ± 0.0291 | 0.5635 ± 0.0218 | 0.5743 ± 0.0213 | 0.5950 ± 0.0203 | **0.6779 ± 0.0325** |
| MT | 0.7521 ± 0.0252 | 0.6757 ± 0.0529 | 0.7720 ± 0.0252 | 0.8139 ± 0.0216 | 0.7868 ± 0.0215 | 0.7929 ± 0.0166 | 0.7798 ± 0.0213 | 0.7915 ± 0.0104 | 0.7693 ± 0.0115 | 0.7845 ± 0.0108 | **0.8180 ± 0.0228** |
| OG | 0.6228 ± 0.0134 | 0.3712 ± 0.1913 | 0.6228 ± 0.0132 | 0.6514 ± 0.0125 | 0.6136 ± 0.0115 | 0.6363 ± 0.0116 | 0.6321 ± 0.0125 | 0.6381 ± 0.0181 | 0.6256 ± 0.0187 | 0.5941 ± 0.0203 | **0.6870 ± 0.0123** |
| PW | 0.9373 ± 0.0315 | 0.9281 ± 0.0161 | 0.9384 ± 0.0152 | 0.9477 ± 0.0136 | 0.9353 ± 0.0137 | 0.9327 ± 0.0296 | 0.9315 ± 0.0152 | 0.8933 ± 0.0053 | 0.9165 ± 0.0050 | 0.8942 ± 0.0053 | **0.9618 ± 0.0143** |
| AD | 0.7992 ± 0.0221 | 0.7645 ± 0.0544 | 0.8115 ± 0.0207 | 0.8199 ± 0.0200 | 0.7875 ± 0.0216 | 0.8019 ± 0.0177 | 0.8006 ± 0.0260 | **0.8247 ± 0.0088** | 0.7620 ± 0.0119 | 0.7810 ± 0.0110 | 0.8200 ± 0.0217 |
| ALB | 0.6065 ± 0.0265 | 0.5512 ± 0.0349 | 0.6196 ± 0.0228 | 0.6494 ± 0.0324 | 0.6096 ± 0.0234 | 0.6354 ± 0.0248 | 0.5929 ± 0.0347 | 0.6047 ± 0.0198 | 0.5997 ± 0.0200 | 0.6192 ± 0.0190 | **0.7038 ± 0.0283** |
| BM | **0.8325 ± 0.0155** | 0.6306 ± 0.1093 | 0.8132 ± 0.0292 | 0.8241 ± 0.0287 | 0.7991 ± 0.0300 | 0.8171 ± 0.0246 | 0.7913 ± 0.0301 | 0.8008 ± 0.0100 | 0.7708 ± 0.0115 | 0.7688 ± 0.0116 | 0.8233 ± 0.0178 |
| CO | 0.4624 ± 0.0086 | 0.4881 ± 0.0250 | 0.5049 ± 0.0172 | 0.5485 ± 0.0160 | 0.5326 ± 0.0167 | 0.4975 ± 0.0111 | 0.4963 ± 0.0143 | 0.5119 ± 0.0244 | 0.5264 ± 0.0237 | 0.5067 ± 0.0247 | **0.5491 ± 0.0150** |
| CC | 0.6169 ± 0.0249 | 0.6302 ± 0.0370 | 0.6218 ± 0.0394 | 0.6451 ± 0.0374 | 0.6780 ± 0.0362 | 0.6690 ± 0.0240 | 0.7190 ± 0.0358 | 0.6408 ± 0.0180 | 0.6616 ± 0.0169 | 0.6247 ± 0.0188 | **0.7331 ± 0.0251** |
| EL | 0.6617 ± 0.0313 | 0.6424 ± 0.0274 | 0.6610 ± 0.0315 | **0.7723 ± 0.0279** | 0.7668 ± 0.0292 | 0.7654 ± 0.0249 | 0.7525 ± 0.0296 | 0.6292 ± 0.0185 | 0.6461 ± 0.0177 | 0.6481 ± 0.0176 | 0.6940 ± 0.0240 |
| HE | 0.4593 ± 0.0119 | 0.3844 ± 0.0119 | 0.4811 ± 0.0119 | — | 0.4590 ± 0.0173 | 0.4853 ± 0.0111 | 0.4448 ± 0.0184 | 0.4465 ± 0.0277 | **0.6477 ± 0.0176** | 0.4854 ± 0.0257 | 0.5462 ± 0.0140 |
| HI | 0.5454 ± 0.0257 | 0.4965 ± 0.0284 | 0.5532 ± 0.0259 | 0.6499 ± 0.0248 | 0.6096 ± 0.0266 | 0.6069 ± 0.0309 | 0.6035 ± 0.0254 | 0.5874 ± 0.0206 | 0.5871 ± 0.0206 | 0.6459 ± 0.0333 | 0.6459 ± 0.0333 |
| JA | 0.5105 ± 0.0166 | 0.4649 ± 0.0292 | 0.5743 ± 0.0248 | **0.5986 ± 0.0254** | 0.5409 ± 0.0262 | 0.5413 ± 0.0159 | 0.5417 ± 0.0249 | 0.4850 ± 0.0258 | 0.4795 ± 0.0260 | 0.4832 ± 0.0258 | 0.5660 ± 0.0212 |
| NU | 0.4833 ± 0.0221 | 0.4932 ± 0.0153 | 0.5221 ± 0.0144 | 0.4333 ± 0.0170 | 0.5308 ± 0.0141 | 0.5231 ± 0.0177 | 0.5517 ± 0.0151 | 0.5900 ± 0.0205 | 0.6296 ± 0.0185 | 0.6775 ± 0.0161 | **0.6545 ± 0.0355** |
| RS | 0.6992 ± 0.0186 | 0.6590 ± 0.0308 | **0.7886 ± 0.0272** | 0.7554 ± 0.0283 | 0.7057 ± 0.0289 | 0.7328 ± 0.0169 | 0.7243 ± 0.0273 | 0.6586 ± 0.0171 | 0.7119 ± 0.0144 | 0.6771 ± 0.0161 | 0.7576 ± 0.0289 |
| VO | 0.4624 ± 0.0086 | 0.3966 ± 0.0340 | 0.4790 ± 0.0306 | 0.5082 ± 0.0313 | 0.4736 ± 0.0304 | 0.4975 ± 0.0111 | 0.4975 ± 0.0111 | 0.4700 ± 0.0265 | 0.4373 ± 0.0281 | 0.5025 ± 0.0249 | 0.5220 ± 0.0138 |
| POL | 0.8515 ± 0.0180 | 0.5000 ± 0.0260 | 0.7480 ± 0.0210 | 0.9520 ± 0.0165 | 0.9425 ± 0.0160 | 0.9330 ± 0.0140 | 0.9321 ± 0.0175 | 0.9150 ± 0.0190 | 0.9070 ± 0.0185 | 0.9280 ± 0.0178 | **0.9538 ± 0.0223** |
| CA | 0.8400 ± 0.0175 | 0.6930 ± 0.0220 | 0.8107 ± 0.0190 | 0.8703 ± 0.0165 | 0.8416 ± 0.0180 | 0.8400 ± 0.0150 | 0.8330 ± 0.0170 | 0.8120 ± 0.0200 | 0.8050 ± 0.0210 | 0.8250 ± 0.0190 | **0.8921 ± 0.0200** |
| EY | 0.4862 ± 0.0190 | 0.3753 ± 0.0250 | 0.5058 ± 0.0180 | 0.5811 ± 0.0200 | 0.5461 ± 0.0160 | 0.5461 ± 0.0140 | 0.5245 ± 0.0150 | 0.5050 ± 0.0180 | 0.4980 ± 0.0170 | 0.5120 ± 0.0165 | **0.5928 ± 0.0190** |
| Mean Accuracy | **0.6380** | 0.5626 | 0.6532 | 0.7012 | 0.6706 | 0.6731 | 0.6721 | 0.6537 | 0.6528 | 0.6546 | **0.7124** |
| Mean Rank | **6.16** | 8.05 | 4.84 | 2.56 | 4.47 | 4.16 | 4.84 | 5.89 | 6.53 | 6.05 | **1.58** |

Table A.2: Ablations classification accuracy at 400 labeled samples (mean ± std).

| Dataset | SS-AE | SS-AE + Gating | OSDT AE + Gating | TANDEM (no gate) | TANDEM (no LRS + Alignment) | TANDEM |
|---|---|---|---|---|---|---|
| CP | 0.6302 ± 0.0318 | 0.6602 ± 0.0318 | 0.6321 ± 0.0303 | 0.6740 ± 0.0324 | 0.6740 ± 0.0324 | **0.6779 ± 0.0325** |
| MT | 0.7796 ± 0.0255 | 0.8096 ± 0.0255 | 0.7639 ± 0.0240 | 0.8117 ± 0.0214 | 0.8014 ± 0.0206 | **0.8180 ± 0.0228** |
| OG | 0.6440 ± 0.0120 | 0.6740 ± 0.0120 | 0.6601 ± 0.0356 | 0.6659 ± 0.0152 | 0.6671 ± 0.0132 | **0.6870 ± 0.0123** |
| PW | 0.9053 ± 0.0181 | 0.9353 ± 0.0181 | 0.8259 ± 0.0310 | 0.9464 ± 0.0129 | 0.9485 ± 0.0141 | **0.9618 ± 0.0143** |
| AD | 0.7785 ± 0.0232 | 0.8085 ± 0.0232 | 0.7688 ± 0.0585 | 0.8111 ± 0.0223 | 0.7990 ± 0.0216 | **0.8200 ± 0.0217** |
| ALB | 0.6497 ± 0.0320 | 0.6797 ± 0.0320 | 0.6652 ± 0.0284 | 0.6763 ± 0.0284 | 0.6800 ± 0.0277 | **0.7038 ± 0.0283** |
| BM | 0.7841 ± 0.0237 | 0.8141 ± 0.0237 | 0.7552 ± 0.0199 | 0.8138 ± 0.0214 | 0.8059 ± 0.0231 | **0.8233 ± 0.0178** |
| CO | 0.4907 ± 0.0170 | 0.5207 ± 0.0170 | 0.5373 ± 0.0499 | 0.5220 ± 0.0148 | 0.5164 ± 0.0163 | **0.5491 ± 0.0150** |
| CC | 0.6600 ± 0.0284 | 0.6900 ± 0.0284 | 0.6892 ± 0.0523 | **0.7434 ± 0.0254** | 0.7188 ± 0.0265 | 0.7331 ± 0.0251 |
| EL | 0.7129 ± 0.0332 | 0.7429 ± 0.0332 | 0.6149 ± 0.0444 | 0.7134 ± 0.0252 | 0.7001 ± 0.0276 | 0.6940 ± 0.0240 |
| HE | 0.5050 ± 0.0136 | 0.5350 ± 0.0136 | 0.4416 ± 0.0656 | 0.5315 ± 0.0139 | 0.5193 ± 0.0142 | **0.5462 ± 0.0140** |
| HI | 0.5979 ± 0.0330 | 0.6279 ± 0.0330 | **0.6621 ± 0.0528** | 0.6300 ± 0.0299 | 0.6350 ± 0.0274 | 0.6459 ± 0.0333 |
| JA | 0.5168 ± 0.0234 | 0.5468 ± 0.0234 | 0.5012 ± 0.0465 | 0.5529 ± 0.0175 | 0.5541 ± 0.0204 | **0.5660 ± 0.0212** |
| NU | 0.5792 ± 0.0342 | 0.6092 ± 0.0342 | 0.6140 ± 0.0393 | 0.6325 ± 0.0318 | 0.6377 ± 0.0307 | **0.6545 ± 0.0355** |
| RS | 0.6716 ± 0.0260 | 0.7016 ± 0.0260 | 0.7252 ± 0.0274 | 0.7311 ± 0.0252 | 0.7335 ± 0.0270 | **0.7576 ± 0.0289** |
| VO | 0.4954 ± 0.0142 | **0.5254 ± 0.0142** | 0.4334 ± 0.0231 | 0.4741 ± 0.0126 | 0.4983 ± 0.0140 | 0.5220 ± 0.0138 |
| POL | 0.9325 ± 0.0220 | 0.9381 ± 0.0210 | 0.9102 ± 0.0300 | 0.9212 ± 0.0195 | 0.9420 ± 0.0185 | **0.9538 ± 0.0230** |
| CA | 0.8653 ± 0.0200 | 0.8393 ± 0.0190 | 0.8493 ± 0.0270 | 0.8721 ± 0.0180 | 0.8803 ± 0.0195 | **0.8921 ± 0.0220** |
| EY | 0.5148 ± 0.0210 | 0.5296 ± 0.0200 | 0.4901 ± 0.0280 | 0.5124 ± 0.0175 | 0.5180 ± 0.0170 | **0.5928 ± 0.0230** |
| Mean Accuracy | **0.6815** | 0.6941 | 0.6600 | 0.6966 | 0.6971 | **0.7124** |
| Mean Rank | **4.45** | 3.61 | 4.71 | 2.92 | 2.79 | **1.74** |

Table A.3: Number of datasets (out of 19; 18 for TabPFN) where TANDEM outperforms each baseline (classification). Parentheses indicate **statistically significant** wins ($p < 0.05$, 100 trials).

| | MLP | XGBoost | TabM | MLogReg | DeepTLF | CatBoost | SS-AE | TabPFN | VIME | SCARF | SubTab |
|---|---|---|---|---|---|---|---|---|---|---|---|
| TANDEM | 18 (17) | 18 (17) | 18 (17) | 19 (18) | 19 (19) | 19 (17) | 18 (18) | 13 (14) | 16 (17) | 17 (19) | 16 (16) |

## A.2    Regression

*Dolan–Moré curves:* Figure 4 in the main paper presents Dolan–Moré performance profiles, which show the proportion of datasets where each model achieves performance within a factor $\tau$ of the best-performing model. Unlike rank-based comparisons, these curves capture both accuracy and robustness, providing a more complete view of model consistency across diverse tasks. Higher curves indicate models that maintain strong performance across a larger share of datasets, even if not ranked first.

Table A.4: Baseline regression MSE at 400 labeled samples (mean ± std).

| Dataset | CatBoost | LogReg | MLP | SCARF | SubTab | VIME | TabM | TabNet | XGBoost | TANDEM |
|---|---|---|---|---|---|---|---|---|---|---|
| BSD | 0.1506 ± 0.0149 | 0.6088 ± 0.0041 | 0.5395 ± 0.0314 | 0.7377 ± 0.0277 | 0.5778 ± 0.0173 | 0.5699 ± 0.0078 | 0.4922 ± 0.0089 | 0.7634 ± 0.2273 | 0.1742 ± 0.0195 | **0.0722 ± 0.0029** |
| BH | 0.0291 ± 0.0010 | 0.0000 ± 0.0000 | 0.0021 ± 0.0007 | 0.0382 ± 0.0045 | 0.0047 ± 0.0006 | 0.0347 ± 0.0018 | 0.0012 ± 0.0004 | 0.0703 ± 0.0000 | 0.0302 ± 0.0015 | **0.0011 ± 0.0002** |
| MBGM | 0.5206 ± 0.0099 | 2.0242 ± 0.4312 | 0.6508 ± 0.0236 | 0.7517 ± 0.0446 | 0.6400 ± 0.0196 | 0.7208 ± 0.0207 | 0.6666 ± 0.0161 | 1.0083 ± 0.1828 | 0.5612 ± 0.0159 | **0.3354 ± 0.0123** |
| SGEM | 0.0340 ± 0.0092 | 0.0041 ± 0.0010 | **0.0017 ± 0.0002** | 0.0038 ± 0.0008 | 0.0113 ± 0.0021 | 0.1848 ± 0.0312 | 0.0032 ± 0.0012 | 0.0231 ± 0.0034 | 0.0171 ± 0.0014 | 0.0025 ± 0.0004 |
| AS | 0.0329 ± 0.0022 | 0.5474 ± 0.0037 | 0.0515 ± 0.0100 | 0.1331 ± 0.0203 | 0.3336 ± 0.0257 | 0.7107 ± 0.0532 | 0.0487 ± 0.0080 | 0.2463 ± 0.0182 | **0.0264 ± 0.0022** | 0.0297 ± 0.0043 |
| BF | **0.6565 ± 0.0335** | 0.8731 ± 0.0096 | 0.8931 ± 0.0221 | 0.8461 ± 0.0265 | 0.8874 ± 0.0208 | 0.9215 ± 0.0472 | 0.8409 ± 0.0230 | 0.9067 ± 0.0385 | 0.6986 ± 0.0466 | 1.0057 ± 0.0316 |
| CO | **0.7047 ± 0.0060** | 0.8370 ± 0.0443 | 0.7711 ± 0.0214 | 0.8030 ± 0.0167 | 0.7869 ± 0.0200 | 0.8152 ± 0.0282 | 0.8055 ± 0.0235 | 0.9399 ± 0.0658 | 0.7395 ± 0.0148 | 0.7334 ± 0.0175 |
| DI | 0.0889 ± 0.0212 | 0.2330 ± 0.1455 | 0.0531 ± 0.0068 | 0.1252 ± 0.0100 | 0.1102 ± 0.0090 | 0.1354 ± 0.0122 | **0.0475 ± 0.0026** | 0.2908 ± 0.0253 | 0.0675 ± 0.0052 | 0.0498 ± 0.0033 |
| EL | **0.1549 ± 0.0058** | 0.5115 ± 0.4341 | 0.2290 ± 0.0262 | 0.2244 ± 0.0213 | 0.2331 ± 0.0171 | 0.1732 ± 0.0096 | 0.1549 ± 0.0058 | 0.2013 ± 0.0141 | 0.1549 ± 0.0058 | 0.1835 ± 0.0079 |
| EY | 0.4987 ± 0.0100 | 0.5237 ± 0.0061 | 0.5533 ± 0.0105 | 0.6110 ± 0.0121 | 0.5825 ± 0.0116 | 0.5591 ± 0.0132 | 0.5577 ± 0.0107 | 0.6253 ± 0.0201 | 0.4805 ± 0.0010 | **0.4143 ± 0.0039** |
| HS | 0.3761 ± 0.0134 | 0.3960 ± 0.0581 | 0.2664 ± 0.0091 | 0.3198 ± 0.0152 | 0.3519 ± 0.0165 | 0.4175 ± 0.0211 | 0.5212 ± 0.0933 | 0.3923 ± 0.0143 | 0.2950 ± 0.0160 | **0.1436 ± 0.0067** |
| VS | **0.0015 ± 0.0002** | 0.1682 ± 0.0025 | 0.0013 ± 0.0001 | 0.0094 ± 0.0016 | 0.0086 ± 0.0013 | 0.3711 ± 0.0586 | 0.0030 ± 0.0010 | 0.0483 ± 0.0082 | 0.0068 ± 0.0014 | 0.0024 ± 0.0006 |
| YP | 1.0557 ± 0.0293 | 2.2639 ± 10.7084 | **1.0278 ± 0.0346** | 1.0804 ± 0.0639 | 1.1325 ± 0.0416 | 1.1236 ± 0.0419 | 1.0419 ± 0.0741 | 1.2104 ± 0.0788 | 1.1751 ± 0.0974 | 1.2300 ± 0.1206 |
| Mean MSE | **0.3318** | 0.6918 | 0.3877 | 0.4372 | 0.4354 | 0.5183 | 0.4006 | 0.5174 | 0.3405 | **0.3234** |
| Mean Rank | **4.00** | 8.38 | 4.38 | 7.23 | 7.08 | 8.46 | 4.85 | 9.38 | 4.15 | **3.38** |

Table A.5: Ablations regression MSE at 400 labeled (MSE mean ± std).

| Dataset | SS-AE | SS-AE + Gating | OSDT AE + Gating | TANDEM (no gate) | TANDEM (no LRS + Alignment) | TANDEM |
|---|---|---|---|---|---|---|
| BSD | 0.0794 ± 0.0040 | 0.0773 ± 0.0039 | 0.0744 ± 0.0037 | 0.0780 ± 0.0039 | 0.0758 ± 0.0038 | **0.0722 ± 0.0036** |
| BH | 0.0012 ± 0.0001 | 0.0012 ± 0.0001 | 0.0011 ± 0.0001 | 0.0012 ± 0.0001 | 0.0012 ± 0.0001 | **0.0011 ± 0.0001** |
| MBGM | 0.3689 ± 0.0184 | 0.3586 ± 0.0179 | 0.3455 ± 0.0173 | 0.3620 ± 0.0181 | 0.3522 ± 0.0176 | **0.3354 ± 0.0168** |
| SGEM | 0.0028 ± 0.0001 | 0.0027 ± 0.0001 | 0.0026 ± 0.0001 | 0.0027 ± 0.0001 | 0.0026 ± 0.0001 | **0.0025 ± 0.0001** |
| AS | 0.0327 ± 0.0016 | 0.0318 ± 0.0016 | 0.0306 ± 0.0015 | 0.0321 ± 0.0016 | 0.0312 ± 0.0016 | **0.0297 ± 0.0015** |
| BF | 1.1063 ± 0.0553 | 1.0751 ± 0.0538 | 1.0369 ± 0.0518 | 1.0862 ± 0.0543 | 1.0560 ± 0.0528 | **1.0057 ± 0.0503** |
| CO | 0.8067 ± 0.0403 | 0.7847 ± 0.0392 | 0.7554 ± 0.0378 | 0.7921 ± 0.0396 | 0.7701 ± 0.0385 | **0.7334 ± 0.0367** |
| DI | 0.0548 ± 0.0027 | 0.0533 ± 0.0027 | 0.0513 ± 0.0026 | 0.0538 ± 0.0027 | 0.0523 ± 0.0026 | **0.0498 ± 0.0025** |
| EL | 0.2019 ± 0.0101 | 0.1962 ± 0.0098 | 0.1890 ± 0.0095 | 0.1982 ± 0.0099 | 0.1927 ± 0.0096 | **0.1835 ± 0.0092** |
| EY | 0.4557 ± 0.0228 | 0.4433 ± 0.0222 | 0.4267 ± 0.0213 | 0.4474 ± 0.0224 | 0.4350 ± 0.0218 | **0.4143 ± 0.0207** |
| HS | 0.1580 ± 0.0079 | 0.1537 ± 0.0077 | 0.1479 ± 0.0074 | 0.1551 ± 0.0078 | 0.1508 ± 0.0075 | **0.1436 ± 0.0072** |
| VS | 0.0026 ± 0.0001 | 0.0026 ± 0.0001 | 0.0025 ± 0.0001 | 0.0026 ± 0.0001 | 0.0025 ± 0.0001 | **0.0024 ± 0.0001** |
| YP | 1.3530 ± 0.0677 | 1.3161 ± 0.0658 | 1.2669 ± 0.0633 | 1.3284 ± 0.0664 | 1.2915 ± 0.0646 | **1.2300 ± 0.0615** |
| Mean MSE | **0.3628** | 0.6551 | 0.9207 | 0.5397 | 0.5832 | **0.3235** |
| Mean Rank | **2.69** | 4.15 | 5.50 | 4.00 | 3.73 | **1.92** |

# B Extended spectral analysis

Extending the spectral analysis in the main paper (Section 6), we present additional visualizations across more datasets (Figure B.1) to further examine how the gating mechanisms in TANDEM shape the frequency content of the input. For each dataset, we focus on a single class and compute the unnormalized discrete Fourier transform (NUDFT) over the 50 most variant features. We compare the spectra of the original input $x$; the neural-gated input $\tilde{x}^{\text{NN}} = x \odot g^{\text{NN}}(x)$, as produced by both TANDEM and SS-AE with gating; and the tree-gated input $\tilde{x}^{\text{OSDT}} = x \odot \bar{g}^{\text{OSDT}}(x)$, where $\bar{g}^{\text{OSDT}}$ is the average gating mask across all trees and depths.

These extended plots confirm the patterns observed in the main paper: neural gating acts as a strong low-pass filter, suppressing high-frequency components. In contrast, tree-based gating preserves more high-frequency variation.

# C Gating activations and dataset profiling

To complement the spectral analysis (Section 5), we present dataset profiling that motivates the choice of datasets used in the gating activation study, followed by summary diagnostics comparing the neural and tree gating behaviors. These diagnostics justify the focused per-dataset gating analysis reported in the Appendix.

## C.1 Dataset profiling: categorical feature ratio

To better understand where TANDEM provides the largest gains, we grouped the benchmark datasets by their ratio of categorical features (i.e., the fraction of categorical input features). The breakdown in Table C.1 was used to select datasets for the detailed gating analyses:

This characterization highlights that TANDEM performs exceptionally well on datasets with a *moderate* share of categorical features, while also showing strong results on fully numeric datasets whose continuous columns have low cardinality (i.e., behave like categorical inputs). The medium categor-

Table A.6: Number of datasets (out of 13) where TANDEM outperforms each baseline (regression). Values in parentheses indicate statistically significant wins (p < 0.05, 100 trials).

| | LogReg | MLP | VIME | SubTab | CatBoost | TabM | SCARF | XGBoost | SS-AE | TabNet |
|---|---|---|---|---|---|---|---|---|---|---|
| TANDEM | 13 (11) | 13 (11) | 13 (10) | 13 (11) | 12 (9) | 11 (10) | 10 (9) | 10 (8) | 9 (9) | 12 (11) |

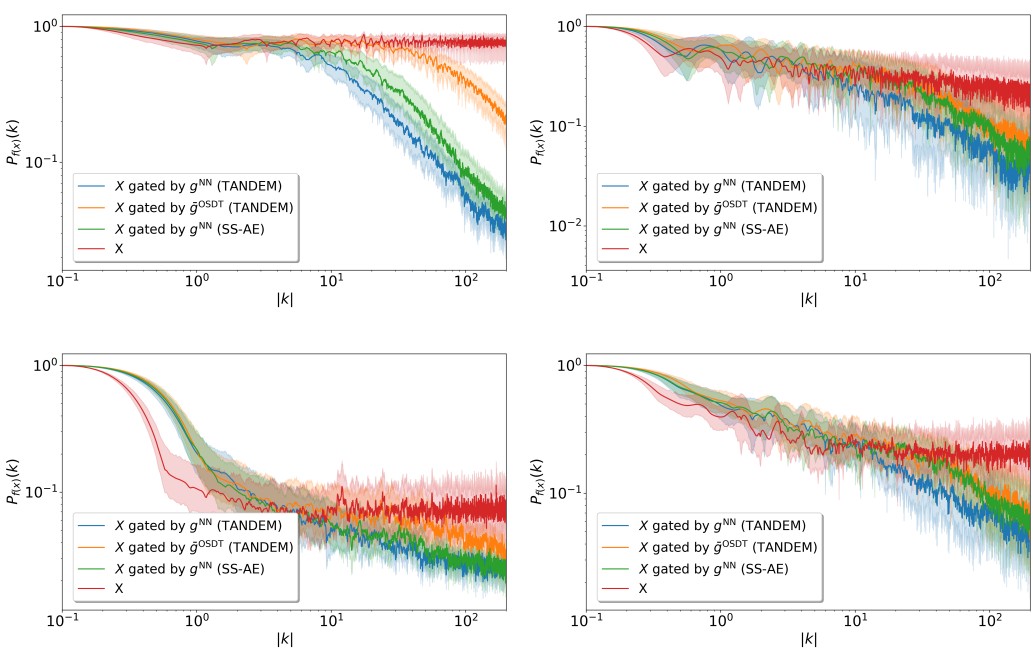

Figure B.1: Frequency spectra of gated inputs for the NN and OSDT encoders. Visualized datasets include OG, CP, VO, and CC. NN gating results in stronger suppression of high-frequency components compared to tree-based gating.

| Category | Datasets (categorical ratio) |
|---|---|
| **High ($\geq 0.70$)** | PW (1.00), BM (0.92), CO (0.85), AD (0.87) |
| **Medium (0.30–0.69)** | RS (0.69), ALB (0.39), EY (0.33), CC (0.33) |
| **Low ($< 0.30$)** | OG (0.10), CP (0.27), VO (0.18), EL (0.12), HI, HE, JA, NU, CA (0.00) |

Table C.1: Dataset grouping by categorical feature ratio used to select datasets for gating analyses.

ical group (RS, ALB, EY, CC), therefore, provides a natural testbed for comparing gating behavior on categorical inputs.

### C.2 Gating diagnostics (selected summaries)

Table C.2 reports interpretable summary statistics computed on the medium-categorical datasets. Definitions: **BinActSim** is cosine similarity between binary activation vectors (i.e., $\mathbf{1}\{\text{gate} > 0.5\}$); **Corr** is the Pearson correlation of gate values; **VarRatio** is the variance of the tree gate divided by the variance of the neural gate, $\dfrac{\text{Var(tree gate)}}{\text{Var(neural gate)}}$; **MeanActOSDT** and **MeanActNN** are the mean gate activations for the tree and neural gating networks, respectively.

Table C.2: Gating similarity and activity statistics (selected datasets).

| Dataset | BinActSim | Corr | VarRatio (OSDT/NN) | MeanActOSDT | MeanActNN |
|---|---|---|---|---|---|
| RS | 0.78 | 0.88 | 1.52 | 0.26 | 0.20 |
| ALB | 0.59 | 0.66 | 1.81 | 0.41 | 0.27 |
| EY | 0.33 | 0.34 | 1.61 | 0.30 | 0.07 |
| CC | 0.65 | 0.82 | 1.76 | 0.43 | 0.37 |

These statistics show that, on the selected medium-categorical datasets, the tree gating (OSDT) tends to be *spikier* (higher variance) and, on average, more active than the neural gating. At the same time, binary overlap and Pearson correlation indicate moderate agreement, suggesting that both encoders consider categorical features necessary.

## C.3 Gating behavior on categorical features

For clarity, Table C.3 repeats the categorical-focused comparison (used in the rebuttal) showing gate overlap, gate correlation, and the variance ratio (Tree/NN) restricted to categorical features only.

Table C.3: Comparison of gating behavior between the neural and tree encoders, evaluated only on the categorical features of four medium-categorical datasets.

| Dataset | Gate Overlap | Gate Correlation | Gate Variance Ratio (Tree / NN) |
|---------|--------------|------------------|----------------------------------|
| RS      | 0.78         | 0.88             | 1.52                             |
| ALB     | 0.59         | 0.66             | 1.81                             |
| EY      | 0.33         | 0.34             | 1.61                             |
| CC      | 0.65         | 0.82             | 1.76                             |

## D  Dataset details

Tables D.1 (classification) and D.2 (regression) summarize the datasets used in our experiments. For classification, we report the number of samples (N), features (F), and target classes (C). For regression, we report the number of samples (N) and features (F).

| Dataset (Acronym) | #Samples (N) | #Features (F) | #Classes (C) |
|-------------------|--------------|---------------|--------------|
| Click_prediction_small (CP) | 9200 | 27 | 2 |
| MagicTelescope (MT) | 19020 | 11 | 2 |
| Otto-Group-Product-Classification-Challenge (OG) | 16400 | 93 | 5 |
| PhishingWebsites (PW) | 9200 | 30 | 2 |
| adult (AD) | 9200 | 14 | 2 |
| albert_categorical (ALB) | 9200 | 25 | 2 |
| bank-marketing (BM) | 45200 | 16 | 2 |
| covertype (CO) | 283300 | 54 | 6 |
| default-of-credit-card-clients_categorical (CC) | 5200 | 23 | 2 |
| electricity (EL) | 19240 | 11 | 2 |
| helena (HE) | 36260 | 27 | 13 |
| higgs (HI) | 470080 | 28 | 2 |
| jannis (JA) | 83600 | 54 | 4 |
| numerai28.6 (NU) | 9200 | 119 | 2 |
| road-safety_categorical (RS) | 363240 | 67 | 2 |
| volkret (VO) | 14800 | 181 | 8 |
| pol (PO) | 15000 | 48 | 2 |
| california (CA) | 20634 | 8 | 2 |
| eye_movements (EM) | 10935 | 27 | 3 |

Table D.1: Dataset statistics (classification): number of samples (N), features (F), and target classes (C).

| Dataset (Acronym) | #Samples (N) | #Features (F) |
|-------------------|--------------|---------------|
| yprop_4_1 (YP) | 7,331 | 251 |
| analcatdata_supreme (AS) | 4,052 | 7 |
| visualizing_soil (VS) | 7,185 | 4 |
| black_friday (BF) | 102,093 | 22 |
| diamonds (DI) | 34,364 | 26 |
| Mercedes_Benz_Greener_Manufacturing (MBGM) | 4,209 | 563 |
| Brazilian_houses (BH) | 8,416 | 18 |
| Bike_Sharing_Demand (BSD) | 12,428 | 20 |
| OnlineNewsPopularity (ONP) | 25,787 | 59 |
| house_sales (HS) | 14,968 | 21 |
| SGEMM_GPU_kernel_performance (SGEM) | 146,960 | 17 |
| electricity (EL) | 29,188 | 8 |
| eye_movements (EY) | 8,562 | 27 |
| covertype (CO) | 350,608 | 54 |

Table D.2: Dataset statistics (regression): number of samples (N) and features (F).

**Note.** For the regression benchmark, we do not require class labels; therefore, dataset selection required only a minimum total of 3,000 (2,000 for pre-training and up to 1,000 for the downstream task) samples per dataset (not 2,500 samples per class as in the classification setting).

# E Compute cost

All timing measurements reported below were collected on an NVIDIA L4 GPU (cloud instance). For completeness, the local workstation used for development and other experiments is: Intel i7 CPU, 16 GB RAM, NVIDIA RTX 3060 GPU (12 GB).

| Model | Mean pretraining time (s) |
|---|---|
| SS-AE | 38.12 |
| TANDEM | 43.47 |
| TabNet | 50.22 |
| SCARF | 63.05 |
| SubTab | 264.89 |
| VIME | 309.76 |

Table E.1: Pretraining time per model (mean across datasets). Measured for 50 pretraining epochs with 2000 samples per label on an L4 GPU.

| Model | Mean downstream time per batch (s) |
|---|---|
| XGBoost | 0.03 |
| MLP | 0.08 |
| TANDEM | 0.08 |
| DeepTFL | 0.10 |
| TabM | 0.12 |
| TabPFN | 0.16 |

Table E.2: Downstream training time per batch (128 samples). Measurements taken on an L4 GPU.

# F Model architectures and training

The architectural components used in our experiments follow the design described in Sections 4.1 and 4.3 of the main paper.

- **Neural Encoder:** 4-layer MLP with BatchNorm and Leaky ReLU activations. Hidden dimensions are chosen to match the embedding size dictated by the OSDT encoder.

- **OSDT Encoder:** Ensemble of oblivious soft decision trees with fixed depth $L$; each tree outputs a soft assignment over $2^L$ leaves. The mean-aggregated output defines the embedding dimension.

- **Neural Decoder:** Mirrors the architecture and dimensionality of the neural encoder.

- **Gating Network:** 2-layer MLP with tanh activation and hard-sigmoid output; used for per-sample feature selection.

- **Fine-tuning MLP:** A single-layer fully connected classifier trained on top of the encoder for downstream supervised evaluation.

# G Hyperparameter tuning

We report mean pretraining time per model (Table E.1) and mean downstream training time per batch (Table E.2).

Table G.1 summarizes the hyperparameter search space used in Optuna-based tuning (50 trials per model, per dataset). Gating components, when applicable, were tuned separately.

# H Optimization robustness

This section reports sensitivity results across key hyperparameters (Table H.1) and optimization cost curves as a function of trial budget (Figure H.1).

Table G.1: Hyperparameter search space for all models. Each model was optimized using Optuna with 50 trials. Gating components (when applicable) were tuned separately.

| Model | Hyperparameter Search Space | Notes |
|---|---|---|
| **TabNet** | learning rate: $[10^{-4}, 10^{-1}]$ 
 mask type: sparsemax, entmax 
 scheduler step size: [5–20], gamma: [0.8–0.95] 
 batch size: 512, 1024, 2048 
 virtual batch size: 64, 128, 256 
 max epochs: [10–100], patience: [5–20] | Pretrained with TabNetPretrainer before supervised fine-tuning |
| **MLP** | learning rate: $[10^{-4}, 10^{-2}]$ (log scale) 
 hidden sizes: [64–256] 
 batch size: 32, 64, 128 
 epochs: [10–50] | 4-layer MLP with ReLU activations |
| **XGBoost** | max depth: [3–10] 
 gamma: $[10^{-8}, 1]$ 
 subsample, colsample bytree: [0.5, 1.0] 
 reg alpha, reg lambda: $[10^{-8}, 10]$ | `use_label_encoder=False` |
| **CatBoost** | depth: [2–10] 
 learning rate: $[10^{-3}, 0.3]$ 
 iterations: [100–300] | `verbose=0` |
| **Logistic Regression** | $C$: $[10^{-4}, 10]$ (log-uniform) | `max_iter=5000` |
| **DeepTFL** | n estimators: [10–100] 
 max depth: [2–7] 
 dropout: [0.0–0.3] 
 n layers: [1–3] | Internal tree-based layers |
| **TabM** | n blocks: [2–4] 
 d block: [64–256] 
 dropout: [0.0–0.3] 
 $k$: [8–64] 
 learning rate: $[10^{-4}, 5 \times 10^{-3}]$ (log) | – |
| **TabPFN** | n estimators: 1, 2, 4, 8 
 softmax temperature: [0.5, 1.0] 
 balance probabilities: True, False 
 average before softmax: True, False | Pretrained; limited tuning allowed |
| **SS-AE** | learning rate: $[10^{-4}, 10^{-2}]$ 
 weight decay: $[10^{-5}, 10^{-2}]$ 
 depth: [3–10] | Encoder uses powers-of-two hidden sizes. Gating tuned separately. |
| **TANDEM** | learning rate: $[10^{-4}, 10^{-2}]$ 
 weight decay: $[10^{-5}, 10^{-2}]$ 
 depth: [3–10] 
 num trees: [2–16] 
 gating optimizer, activation, learning rate | Shared decoder. Gating network tuned separately. |
| **Gating Network** | hidden size: [32–128] 
 learning rate: $[10^{-4}, 10^{-2}]$ 
 weight decay: $[10^{-5}, 10^{-2}]$ | Used in SS-AE and TANDEM for per-sample masking. |

## H.1 Sensitivity analysis

In order to assess model robustness, we vary key hyperparameters: alignment loss weight $\lambda_{\text{align}}$, learning rate, embedding dimension, number of trees, and gate temperature across five representative datasets (CP, OG, PW, ALB, CC). As shown in Table H.1, the best performance is maintained across a wide range of values, indicating that TANDEM is robust and not sensitive to narrow hyperparameter settings.

| Parameter | Value | CP | OG | PW | ALB | CC |
|---|---|---|---|---|---|---|
| **Alignment loss weight** ($\lambda_{\text{align}}$) | | | | | | |
| | 0.0 | 0.6702 | 0.6721 | 0.9332 | 0.6790 | 0.7165 |
| | 0.1 | 0.6732 | 0.6789 | 0.9387 | 0.6781 | 0.7286 |
| | 0.5 | 0.6775 | 0.6865 | 0.9613 | 0.7032 | 0.7326 |
| | 1.0 | 0.6753 | 0.6860 | 0.9620 | 0.7038 | 0.7231 |
| | 10.0 | 0.6105 | 0.6230 | 0.8930 | 0.6211 | 0.6815 |
| **Learning rate** (`lr`) — mean $\pm$ std | | | | | | |
| | 1e-4 | $0.6582 \pm 0.012$ | $0.6380 \pm 0.011$ | $0.9521 \pm 0.010$ | $0.6912 \pm 0.015$ | $0.7201 \pm 0.013$ |
| | 5e-4 | $0.6705 \pm 0.013$ | $0.6485 \pm 0.012$ | $0.9587 \pm 0.009$ | $0.6975 \pm 0.014$ | $0.7268 \pm 0.012$ |
| | 1e-3 | $0.6775 \pm 0.014$ | $0.6560 \pm 0.013$ | $0.9613 \pm 0.008$ | $0.7032 \pm 0.013$ | $0.7326 \pm 0.012$ |
| | 5e-3 | $0.6740 \pm 0.016$ | $0.6472 \pm 0.017$ | $0.9502 \pm 0.012$ | $0.6890 \pm 0.018$ | $0.7190 \pm 0.015$ |
| | 1e-2 | $0.6601 \pm 0.018$ | $0.6305 \pm 0.020$ | $0.9310 \pm 0.020$ | $0.6723 \pm 0.021$ | $0.7015 \pm 0.019$ |
| **Embedding dimension** (mean $\pm$ std) | | | | | | |
| | 8 | $0.6681 \pm 0.012$ | $0.6510 \pm 0.014$ | $0.9562 \pm 0.010$ | $0.6950 \pm 0.014$ | $0.7280 \pm 0.013$ |
| | 16 | $0.6724 \pm 0.013$ | $0.6548 \pm 0.013$ | $0.9591 \pm 0.009$ | $0.6979 \pm 0.013$ | $0.7302 \pm 0.012$ |
| | 32 | $0.6775 \pm 0.014$ | $0.6560 \pm 0.013$ | $0.9613 \pm 0.008$ | $0.7032 \pm 0.013$ | $0.7326 \pm 0.012$ |
| | 64 | $0.6760 \pm 0.015$ | $0.6550 \pm 0.015$ | $0.9600 \pm 0.010$ | $0.7020 \pm 0.015$ | $0.7310 \pm 0.014$ |
| **Number of trees** (# trees) — mean $\pm$ std | | | | | | |
| | 50 | $0.6680 \pm 0.013$ | $0.6480 \pm 0.014$ | $0.9580 \pm 0.009$ | $0.6950 \pm 0.013$ | $0.7288 \pm 0.012$ |
| | 100 | $0.6738 \pm 0.013$ | $0.6525 \pm 0.013$ | $0.9609 \pm 0.009$ | $0.6990 \pm 0.012$ | $0.7312 \pm 0.012$ |
| | 200 | $0.6775 \pm 0.014$ | $0.6560 \pm 0.013$ | $0.9613 \pm 0.008$ | $0.7032 \pm 0.013$ | $0.7326 \pm 0.012$ |
| | 500 | $0.6764 \pm 0.015$ | $0.6558 \pm 0.015$ | $0.9610 \pm 0.010$ | $0.7025 \pm 0.015$ | $0.7320 \pm 0.014$ |
| **Gate temperature** (softmax temperature) | | | | | | |
| | 0.1 | $0.6720 \pm 0.013$ | $0.6580 \pm 0.014$ | $0.9608 \pm 0.009$ | $0.7001 \pm 0.013$ | $0.7302 \pm 0.012$ |
| | 0.5 | $0.6765 \pm 0.013$ | $0.6555 \pm 0.013$ | $0.9612 \pm 0.008$ | $0.7030 \pm 0.013$ | $0.7320 \pm 0.012$ |
| | 1.0 | $0.6775 \pm 0.014$ | $0.6560 \pm 0.013$ | $0.9613 \pm 0.008$ | $0.7032 \pm 0.013$ | $0.7326 \pm 0.012$ |
| | 2.0 | $0.6740 \pm 0.015$ | $0.6520 \pm 0.015$ | $0.9590 \pm 0.011$ | $0.7000 \pm 0.015$ | $0.7290 \pm 0.014$ |

Table H.1: Concatenated sensitivity results for selected hyperparameters and datasets (CP, OG, PW, ALB, CC). Values are classification accuracy; entries marked "mean $\pm$ std" show the mean and standard deviation across runs.

## H.2 Cost-based optimization (trial budget analysis)

To empirically assess optimization robustness, we ran Optuna hyperparameter searches and measured performance as a function of trial budget. Figure H.1 reports mean accuracy and standard deviation computed over a moving window of the last five trials at each budget size, across different random seeds and hyperparameter samples.

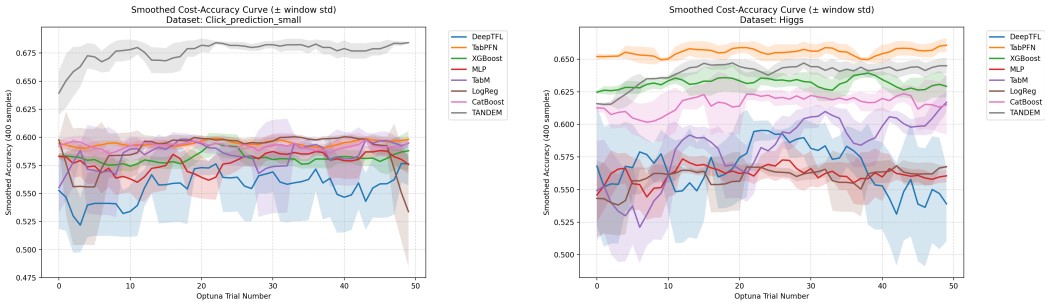

Figure H.1: Mean accuracy over trials (moving window of last 5 trials) on the CP (left) and HI (right) datasets as a function of trial budget. Curves show model mean accuracy, and shaded regions indicate the corresponding standard deviation across seeds/hyperparameter samples.

