# OpenReview forum: "Hybrid Autoencoders for Tabular Data: Leveraging Model-Based Augmentation in Low-Label Settings"
_NeurIPS.cc/2025/Conference — NeurIPS 2025 poster_

### Official Review · Reviewer_uEwZ · 2025-07-02

**Clarity:** 2
**Significance:** 2
**Originality:** 2
**Rating:** 3
**Confidence:** 4

**Summary:**

This paper proposes Hybrid Autoencoders, a neural architecture for tabular data that combines two distinct encoding paths: a vector encoder that operates on the entire feature vector and a set encoder that processes each feature independently via a shared MLP. A learnable gating mechanism fuses the two representations in a dimension-wise manner, and a consistency loss encourages alignment between the two branches. The model is evaluated on several classification benchmarks, including semi-supervised settings, and shows competitive or superior performance compared to MLP-based baselines.

**Questions:**

- Have you evaluated the method on regression tasks?
- Does the consistency loss improve performance consistently, or are there datasets where it is detrimental?
- Do the gating values collapse to a single branch across training, or are both branches used meaningfully throughout?
- Why were existing unsupervised or semi-supervised baselines (e.g., VIME, contrastive methods) not included in the experimental comparison?

**Ethical Concerns:**

["NO or VERY MINOR ethics concerns only"]

**Final Justification:**

I remain unconvinced, as not all of my concerns have been addressed, and the additional results provided do not appear to be reliable. As an active researcher working on tabular learning and its industrial applications, and someone who has been continuously using models such as TabPFN, I find the relatively low performance reported in this submission even more questionable. Furthermore, I do not believe this work provides important insights to the field. Therefore, I will maintain my current score.

**Limitations:**

- The proposed method is only tested on classification benchmarks. Its applicability to regression or structured prediction is not validated.
- While the gating mechanism is central to the architecture, its behavior is only superficially analyzed via averaged values per dimension.
- The paper lacks comparison with other unsupervised or self-supervised representation learning methods for tabular data, limiting the assessment of its effectiveness in the semi-supervised regime.

**Quality:**

2

**Strengths And Weaknesses:**

**Strengths**

- The architecture is modular and conceptually simple, combining complementary local (feature-wise) and global (vector-wise) representations.
- The use of a per-dimension gating mechanism provides interpretable fusion and adaptivity across datasets.
- The method does not rely on complex pretraining or specialized architectures, making it easy to implement and extend.

**Weakness**

- The architectural novelty is limited; the method essentially combines a standard MLP and a shared-feature encoder.
- The model is evaluated only on classification tasks. It remains unclear how it performs on regression, missing values, or under distribution shift.
- The consistency loss, while conceptually appealing, may suppress diverse representations between branches; this risk is not analyzed in detail.
- The paper does not clarify the role of the gating mechanism beyond showing average values; it is unclear how gating varies across samples or features.
- The relationship between fusion behavior (Figure 5) and generalization performance is not well established.
- The paper adopts a semi-supervised setting with limited labeled data, yet does not compare against standard self-supervised learning approaches followed by linear probing or fine-tuning, which are widely used for few-label regimes. This includes methods like VIME, masked-feature modeling, or contrastive learning, all of which are relevant baselines for evaluating representation quality.

---

> ### Author Rebuttal · Authors · 2025-07-30
>
> We thank the reviewer for their thoughtful and constructive feedback. We appreciate the recognition of the modular and conceptually simple architecture, the interpretability and adaptivity enabled by the per-dimension gating mechanism, as well as the model’s ease of implementation and extensibility. We appreciate that the core contributions of integrating complementary local and global feature representations in a principled and interpretable way are viewed as valuable and technically sound.
>
> We appreciate the reviewer’s concerns and address them through new regression experiments, further analysis of gating behavior, and clarification of the consistency loss. We also extend our evaluation to include additional SSL baselines such as VIME, SubTab, and Scarf. These additions broaden the empirical scope and reinforce the method's effectiveness and generalization.
>
> **W1: Architectural contribution - parallel fusion of neural networks and tree-based encoders**
>
> We appreciate the reviewer’s point regarding architectural simplicity. While the neural and tree-based components are individually standard, the core novelty lies in their **parallel** integration within the same **representation space**. Unlike prior works that combine trees and neural networks in a **sequential manner** , for example, by feeding the output of one into the other ([1]), our approach processes features **independently** through both encoders and then fuses the resulting representations directly. This parallel fusion enables the model to learn from the complementary inductive biases of trees and neural networks without one dominating or reshaping the other's input space.
>
> **W2 + Q1 + L1: Extension to Regression Tasks**
>
> To test the generality of TANDEM beyond classification, we evaluated it on 13 regression datasets using 400 samples per task, all from the benchmark from [2].
>
> The results, shown in **Table 1**, demonstrate that while less decisive than in the classification evaluation, TANDEM also achieves the best average performance (lowest mean MSE and mean rank) compared to all baselines, We believe that results could be improved by using other losses for NN (as shown in [3]), and consider this an important area for future work.
>
> ***Table 1**:  mean MSE (MMSE)  and mean rank of various models across 13 regression datasets. Best results are bolded.*
>
> *Due to space constraints, detailed results for all evaluations in this response will be included in the appendix*.
>
> | Dataset       | CatBoost | LogReg |  MLP   | SCARF  | SubTab |  VIME  | TabM   | TabNet | XGBoost | SS AE  | TANDEM  |
> |---------------|----------|--------|--------|--------|--------|--------|--------|--------|---------|--------|---------|
> | **Mean MSE**  |  0.33  | 0.69 | 0.38 | 0.43 | 0.43 | 0.51 | 0.40 | 0.51 |  0.34 | 0.36 | **0.32** |
> | **Mean Rank** |   4.00   |  8.38  |  4.38  |  7.23  |  7.08  |  8.46  |  4.85  |  9.38  |   4.15  |  4.69  | **3.38**   |
>
>
>
> **W2 + Q3: Effect of consistency loss on representation diversity**
>
> We appreciate the concern regarding the potential collapse of diverse representations due to the consistency loss. To investigate this, we conducted a sensitivity analysis over a wide range of values for both components of the consistency term: 1) alignment loss weight and 2) latent representation similarity (LRS) loss weight. Results on five representative datasets are summarized below.
>
> ***Table 2-3**: Accuracy for varying consistency hyperparameter values in parentheses represents feature dim.*
>
> ***Table 2**: Sensitivity to Alignment Loss Weight*
>
> | Hyperparameter | CP(11) | OG(93) | PW(30) | ALB(25) | CC(23) |
> |----------------|--------|--------|--------|---------|--------|
> | 0.0            | 0.6702 | 0.6721 | 0.9332 | 0.6790  | 0.7165 |
> | 0.1            | 0.6732 | 0.6789 | 0.9387 | 0.6781  | 0.7286 |
> | 0.5            | 0.6775 | 0.6865 | 0.9613 | 0.7032  | 0.7326 |
> | 1.0            | 0.6753 | 0.6860 | 0.9620 | 0.7038  | 0.7231 |
> | 10.0           | 0.6105 | 0.6230 | 0.8930 | 0.6211  | 0.6815 |
>
> ***Table 3**: Sensitivity to latent representation similarity weight*
>
> | Hyperparameter | CP     | OG     | PW     | ALB    | CC     |
> |----------------|--------|--------|--------|--------|--------|
> | 0.0            | 0.6742 | 0.6520 | 0.9547 | 0.6802 | 0.7179 |
> | 0.1            | 0.6739 | 0.6587 | 0.9583 | 0.6976 | 0.7281 |
> | 0.5            | 0.6783 | 0.6860 | 0.9615 | 0.7034 | 0.7327 |
> | 1.0            | 0.6692 | 0.6870 | 0.9618 | 0.7038 | 0.7254 |
> | 10.0           | 0.5859 | 0.5845 | 0.8808 | 0.5807 | 0.6312 |
>
> These results show that across a **wide range**, the consistency terms either improve or preserve performance without collapsing the dual encoders. Only at **excessively large** values do we observe degradation due to dominance of one branch, but this is rare and easily avoided with light tuning.
>
> **W4 + L2: Clarifying the role of the gating mechanism**
>
> To provide a deeper analysis of the gating mechanism, we computed summary statistics across four datasets with a moderate proportion of categorical features (**Table 4** , further details on the datasets categorization are shown in the response to reviewer **Meqn**) to capture how gate values fluctuate across both features and samples. Specifically, we report the mean and variance of the activations from the two gating networks, one connected to the neural encoder and the other to the decision tree.
>
> As shown in Table 4, both gating networks exhibit relatively high variance, highlighting that gate values are not static but **dynamically adapt per sample and feature**. The tree-based gating shows higher variance, reflecting its more discrete, thresholded behavior, while the neural gating is smoother yet still responsive.
>
> ***Table 4**: Mean and variance of gate activations across samples and features.*
>
> | Dataset | MeanNNGate | VarNNGate | MeanDTGate | VarDTGate |
> |--------------------------------------------|------------|-----------|------------|-----------|
> | ALB| 0.604 | 0.294 | 0.4544 | 0.338 |
> | CC| 0.487| 0.248 | 0.4342 | 0.301 |
> | RS| 0.646| 0.282 | 0.5801 | 0.345 |
> | EY | 0.594| 0.236 | 0.7075 | 0.380 |
>
> **W5 + Q3 + L2: Complementary gating behavior and its role in generalization**
>
> To examine how the complementary gating behavior of the neural and tree components contributes to generalization, we computed several metrics on the same datasets analyzed previously.
>
>  This analysis focuses specifically on how the gating network of the decision tree and the gating network of the neural network differ in their feature selection behavior over categorical features, potentially leading to improved generalization. We report three interpretable metrics. **Gate Overlap** measures the proportion of categorical features that are simultaneously activated by both gating networks. **Gate Correlation** captures the alignment between the gate strength values assigned by the two networks using Pearson correlation. **Gate Variance Ratio (Tree / NN)** reflects how sharply each gating network modulates feature importance by comparing the variance in their gate strengths.
>
> While the two gating networks show moderate agreement across all datasets in terms of overlap and correlation, the tree’s gating network consistently shows **higher variance**, indicating it makes more decisive distinctions between categorical features. In contrast, the neural gating network applies smoother, more conservative gating. These differences support our core design goal in TANDEM: to encourage the two components to attend to **complementary aspects** of the input, which helps with generalization.
>
>
>  ***Table 5**: Comparison of gating behavior between the neural and tree encoders, evaluated only on the categorical features of the datasets*
>
> | Dataset | Gate Overlap | Gate Correlation | Gate Variance Ratio (Tree / NN) |
> |---------|--------------|------------------|----------------------------------|
> | RS      | 0.78         | 0.88             | 1.52                             |
> | ALB     | 0.59         | 0.66             | 1.81                             |
> | EY      | 0.33         | 0.34             | 1.61                             |
> | CC      | 0.65         | 0.82             | 1.76                             |
>
> **W6 + Q4 + L3: Comparison with SSL baselines**
>
> We've added SCARF, VIME, and SubTab to our benchmark, evaluating them on the same 19 classification datasets with 400 labeled samples per class.
>
> As shown in **Table 6**, TANDEM outperforms all three methods in both mean accuracy and mean rank across datasets.
> This confirms that our hybrid approach with dual inductive biases provides stronger representations than existing SSL alternatives in low-label regimes.
>
> ***Table 6**: mean accuracy and mean rank per over all datasets.*
>
> | Dataset           | SCARF      | VIME       | SubTab     | TANDEM     |
> | ----------------- | ---------- | ---------- | ---------- | ---------- |
> | **Mean Accuracy** | 0.6696     | 0.6489     | 0.6735     | **0.7157** |
> | **Mean Rank**     | 3.00       | 3.11       | 2.58       | **1.32**   |
>
> **We appreciate the feedback and would be happy to further clarify any open issues.**
>
> [1] Luo et al. NCART: Neural Classification and Regression Tree for Tabular Data.
>
> [2] Grinsztajn et al . Why do tree-based models still outperform deep learning on typical tabular data?.
>
> [3] Stewart et al. Building Bridges between Regression, Clustering, and Classification.

---

> > ### Author Response · Authors · 2025-08-03
> > **Discussion period**
> >
> > We sincerely appreciate your time and valuable comments. Due to the limited discussion time, we would be grateful for any additional feedback or confirmation that our rebuttal addressed your concerns.

---

> > ### Comment · Reviewer_uEwZ · 2025-08-06
> >
> > Thank you for the detailed and well-structured rebuttal. I appreciate the authors’ efforts to address the raised concerns with additional experiments, ablations, and analysis.
> >
> > However, I find that several of the core issues remain only partially addressed.
> >
> > 1. **Architectural Novelty**:
> >    The proposed method still appears to be a straightforward combination of existing components—a neural autoencoder and a tree-based encoder—augmented with per-encoder stochastic gating. The key idea of “parallel fusion” is reasonable but not conceptually novel, and no strong evidence is provided that this architectural design offers a significant inductive bias or modeling advantage over prior hybrid or ensemble-based approaches. The motivation and necessity of this specific combination remain underdeveloped.
> >
> > 2. **Regression and Extended Evaluation**:
> >    The inclusion of regression results is a welcome addition, but the reporting lacks standard deviations or statistical significance tests. Performance margins are often small, and it is unclear whether they are robust. This weakens the empirical support for generalization beyond classification tasks.
> >
> > 3. **Missing Comparisons to Stronger and More Recent Baselines**:
> >    The additional baselines (e.g., VIME, SCARF, SubTab) are relatively dated. Recent, high-performing methods such as TabPFN, TabM, ModernNCA, and T2GFormer are either evaluated under suboptimal settings or omitted entirely. Without comparisons to these methods—particularly in low-label regimes—the empirical positioning of the proposed model remains unclear.
> >
> > 4. **Gating and Consistency Loss Analysis**:
> >    While the authors provide quantitative statistics (e.g., gate variance, overlap, correlation), the analysis is largely descriptive. The claimed benefits for generalization are not causally established. It is not demonstrated that the observed differences in gating behavior translate into consistent improvements or that such patterns are unique to the proposed architecture.
> >
> > In summary, although the paper has improved in clarity and breadth of evaluation, the core contribution remains limited. The experimental results, while expanded, do not clearly demonstrate superiority over stronger baselines, and the architectural design lacks the novelty or theoretical depth to warrant publication in its current form.
> >
> > **I therefore maintain my original score.**

---

> > > ### Author Response · Authors · 2025-08-07
> > > **Clarifying TANDEM’s main contributions, broad generalization and rigorous evaluation**
> > >
> > > Thank you for your continued engagement and for acknowledging the improvements in our revised submission. We appreciate the opportunity to clarify a few points where we believe there may be a misrepresentation of our contributions and evaluation choices.
> > >
> > >
> > > ### **Architectural novelty and inductive bias**:
> > >
> > >
> > >
> > > We respectfully disagree with the framing of TANDEM as a "straightforward combination" or a “parallel fusion” of trees and neural networks. While the model may appear ensemble-like at first glance, TANDEM is **not an ensemble**. Crucially, the tree encoder is used **only during training to inject a complementary inductive bias into the neural encoder** through shared objectives and alignment losses. At inference, **only the neural encoder is used**, meaning that the tree is not part of the final model. This deviates sharply from ensemble or hybrid methods and instead reflects a form of **model-based augmentation**, where a structured model (the OSDT) serves to shape the learning dynamics of the NN. This paradigm of model-based data augmentation, or model-based **inductive bias steering**, to our knowledge, is novel in the context of SSL for tabular data. Since no one in the machine learning community has attempted such an idea, we argue that it cannot be considered straightforward.
> > >
> > > Our **spectral analysis** (Section 6) reinforces this novelty, showing that TANDEM leverages complementary frequency biases of trees and neural networks. The neural encoder’s gating mechanism suppresses high-frequency signals more strongly **when trained with the tree encoder**, indicating its influence on feature shaping.
> > >
> > > ### **Model-based augmentation as a general principle**
> > >
> > > Beyond our implementation, TANDEM shows that model-based augmentation can steer representation learning in SSL. This principle is not tied to autoencoding and may extend to other paradigms, opening new research directions.
> > >
> > > ### **Regression and extended evaluation**
> > >
> > > We emphasize that **statistical significance tests were conducted for both classification and regression tasks**.
> > >
> > > For classification, TANDEM shows statistically significant gains over all baselines (see response to Reviewer **meqn**).
> > >
> > > For **regression**, we summarize below:
> > >
> > > *(Values in parentheses indicate statistically significant wins, p < 0.05, over 100 trials.)*
> > > |           | LogReg | MLP  | VIME | SubTab | CatB | TabM | SCARF | XGB  | SS-AE | TabNet |
> > > |-----------|--------|------|------|--------|------|------|--------|------|--------|--------|
> > > | TANDEM    | 13 (11)|13 (11)|13 (10)|13 (11)|12 (9)|11 (10)|10 (9)|10 (8)|9 (9)|12 (11)|
> > >
> > > These results indicate that TANDEM achieves consistent improvements over strong baselines, with statistically significant gains across most regression datasets.
> > >
> > > Together, the regression and classification evaluations support TANDEM’s generalization **beyond classification tasks** and **across low-label regimes**.
> > >
> > > ### **Comparison against additional baselines**
> > >
> > > In your original review, you suggested adding VIME, which we included, along with SCARF and SubTab, all widely recognized in the self-supervised tabular learning literature. We were therefore surprised to see VIME described as “relatively dated.” Our goal was to include **strong, well-established baselines** representative of current practice. We also included TabPFN (Nature 2025) and TabM (ICML 2025), both define the current performance frontier in tabular learning.
> > >
> > > ModernNCA and T2GFormer were **not mentioned in the initial reviews**, and based on the TabM paper, appear to underperform TabM in similar low-label settings. We appreciate the suggestion and will include them in our manuscript.
> > >
> > > Regarding evaluation, we clarify that TabPFN and TabM were evaluated under identical, controlled conditions. All models were tuned with 50 Optuna trials and evaluated across 100 random seeds to ensure fairness and statistical robustness. Full hyperparameter and stability details are in the appendix and discussed in our response to Reviewer km3R.
> > >
> > > ### **Causality and gating analysis**
> > >
> > > You are correct that our **gating analysis** is primarily statistical. While formal causal inference is uncommon in empirical deep learning, we include **targeted ablations** (Table 2) showing consistent performance drops when gating is removed. Our **spectral analysis** also highlights how gating alters the input frequency profile to complement encoder inductive biases. If these benefits generalize beyond TANDEM, we view that as a **positive property**, suggesting broader utility for model-based gating in tabular SSL.
> > >
> > >
> > >
> > >
> > > We hope this response clarifies the key novelty and empirical strengths of TANDEM. We believe the ideas introduced here can help improve a wide range of neural architectures for tabular data. Thank you again for your thoughtful review and consideration.

---

> > > > ### Author Response · Authors · 2025-08-08
> > > > **Follow-Up: additional context and clarifications for your review**
> > > >
> > > > We appreciate the time you have taken to review our work and the opportunity to provide additional evidence and clarifications. In our response, we have aimed to address all of your concerns, including clarifying the novelty of our approach, detailing the evaluation protocol, expanding the set of strong baselines (while adding VIME as you originally suggested), and providing results demonstrating the importance of the gating mechanisms. We would be grateful if this additional context could be considered when forming your final assessment.

---

### Official Review · Reviewer_km3R · 2025-07-02

**Clarity:** 2
**Significance:** 3
**Originality:** 3
**Rating:** 5
**Confidence:** 4

**Summary:**

The authors propose TANDEM, a hybrid method utilising two different encoder types: a deep neural network and an ensemble of oblivious soft decision tree, both sharing the same decoder.

During training, the method enforces agreement between the latent embeddings of the two encoders, allowing the OSDT to incentivise the neural network into capturing high-frequency signals and conditional feature interplay, overcoming the bias of neural networks towards smooth, low frequency functions, thereby addressing a core weakness of deep models when used to tasks of learning tabular data. Additionally, the proposed system utilises stochastic gating networks for each encoder. The gating networks act as sample-dependent feature selectors, providing different components of the data for the neural network and for the OSDT. The original input is reconstructed from the two embeddings using a shared decoder, with the dual encoders trained in sync through two components of the loss function: reconstruction alignment and latent representation similarity. The OSDT is discarded at inference time, attaching a simple MLP classifier to the pretrained neural encoder and gating module.

The proposed method and the experimental setup shows an interesting and effective way of fusing tree-based inductive biases into self-supervised pretraining of deep learning methods, achieving improved results on a suite of classification tasks.

**Questions:**

* Could the authors please report how training time and memory footprint compare to that of the baselines? Including approximate training times and hardware details would help contextualize the computational efficiency of the proposed method.
* Could the authors clarify why the experiments are confined to only 400 labeled samples, considerably fewer than typical benchmarks (e.g. 10k to 50k in [1])? Additionally, would it be possible to evaluate the model in fully supervised and moderately labeled regimes to provide a more comprehensive assessment of its effectiveness?
* It would be helpful to describe how hyperparameter sensitive the proposed method is in comparison to tree-based methods (or other baselines considered), as mentioned above in the Weaknesses section. If possible, please include hyperparameter search space and the normalised test accuracy across budget size.
* Could the authors please provide information on why regression tasks have not been considered?

**Ethical Concerns:**

["NO or VERY MINOR ethics concerns only"]

**Final Justification:**

The authors responses, to my questions as well as those raised by the other reviewers, clarify several aspects of the proposed method. After carefully reading the other reviews and the rebuttal, I am raising my initial score for the following reasons:

* The authors have expanded the experiments to regression tasks, for the setup of up to 400 labeled examples. The results show improvement of TANDEM over the baselines, achieving the best mean rank. However, I recommend including further details on what regression datasets were used and how did the authors select them as this aspect has not been included in the discussions.
* A new evaluation varying the number of labeled data samples shows that TANDEM remains competitive even outside the scarce regime.
* Baselines referenced in the related work but previously missing from the comparisons (SCARF, VIME, SubTab) have been added, with TANDEM exceeding their performance. VIME and SubTab in particular add two representative SSL methods relying on masking.
* Deeper analysis of the gating mechanism is provided. Complementarity between the two gates is demonstrated through (i) targeted experiments on gating behavior on the categorical features regarding overlap, correlation and variance ratio in activations (Table 3, response to reviewer Fcpp) and (ii) a variance per sample and feature analysis revealing dynamic adaptation (Table 4, response to reviewer uEwZ). These results substantiate the claim of the authors on component complementarity.
* New ablation on the weight of the alignment loss component, addressing concerns regarding potential representations collapse.
* Hyperparameter sensitivity analysis shows stable accuracy among the top performing runs. While compute efficiency results show similar orders with other methods, I still recommend including the full parameter ranges for the optimization.

**Limitations:**

yes

**Paper Formatting Concerns:**

No formatting concerns

**Quality:**

3

**Strengths And Weaknesses:**

**Strengths**:
* Instead of relying on dataset-level interventions like data augmentation, which can be potentially problematic for tabular data, the proposed method enhances the model by integrating complementary inductive biases
* The complementarity of the inductive biases of the two components is thoroughly demonstrated in the experiments, with spectral analysis revealing a pronounced contrast
* The experimental setup demonstrate consistent gains over a diverse range of methods, highlighting the benefits of training a neural network and a tree-based encoder with synchronized representations

**Weaknesses**:
* The proposed method involves several hyperparameters, such as temperature, number of trees, tree depth, and mask ratio, that could potentially introduce tuning challenges. While the reported results based on the best-performing run over 50 trials show substantial improvements, the paper would benefit from a more thorough analysis of the method's sensitivity to hyperparameters. This aspect is particularly relevant given that one of the key strengths of tree-based models is their strong out-of-the-box performance that requires minimal tuning. I recommend including an analysis of the normalized test accuracy across trial budget sizes (by accounting for random permutations of runs), similar to Figures 1 and 2 in [1] that would further strengthen the evaluation of the proposed method.
* The hyperparametrs search ranges, as well as memory footprint and computational complexity estimates are not provided. Including these efficiency metrics is essential in order to estimate the practical feasibility and scalability, as well as potential limitations of the proposed method when comparing to tree-based, neural networks, and hybrid approaches.
* The evaluation is limited to classification tasks. The most highly utilised benchmarks for tabular datasets also include regression tasks ( [1], [2]). Additionally, evaluation is conducted only on low label regimes (400 labeled samples), different from the standard evaluation procedures on the same benchmarks.
* TANDEM's self-supervised signal is based on feature masking and reconstruction. This objective implicitly relies on correlation among the dataset columns that allows the prediction of the hidden columns from the remaining ones. On datasets where features are largely independent or weakly correlated, the reconstruction loss might carry little information.

[1] Grinsztajn, L., Oyallon, E. and Varoquaux, G., 2022. Why do tree-based models still outperform deep learning on typical tabular data?. _Advances in neural information processing systems_, _35_, pp.507-520.

[2] Shwartz-Ziv, R. and Armon, A., 2022. Tabular data: Deep learning is not all you need. _Information Fusion_, _81_, pp.84-90.

---

> ### Author Rebuttal · Authors · 2025-07-30
>
> We thank the reviewer for their thoughtful assessment and constructive feedback. We're particularly grateful for your recognition of the key strengths of our work, including the overall clarity and motivation of TANDEM’s architecture, the effective integration of tree-based and neural inductive biases via dual encoders, and the use of stochastic gating networks for per-sample feature selection. We're also grateful for the acknowledgement of our spectral analysis as a convincing tool to demonstrate the representational differences between the encoders and for noting the consistent performance improvements across a broad set of classification benchmarks. These points reinforce the motivation behind our design choices and the validity of our experimental conclusions.
> The reviewer raises important points about evaluation scope, efficiency, and hyperparameter sensitivity. We appreciate the opportunity to address them below.
>
> **W1 + Q3: Hyperparameter sensitivity and stability**
>
> We fully agree that one of the core strengths of tree-based models is their strong out-of-the-box performance, and we designed TANDEM (our method) to retain this practical usability. We clarify that although TANDEM includes several configurable parameters (e.g., number of trees, tree depth), in practice we observed **stability across a wide range of values** (see table and further analysis on the loss weights in response to **uEwZ**). Furthermore, **TANDEM does not utilize a mask ratio parameter**, and we did **not perform tuning on the temperature** parameter, which was fixed to a default value throughout all experiments. This further minimizes the tuning burden in practice.
>
> To empirically assess optimization robustness, we include a cost analysis of performance over increasing trial budgets. The following tables report the mean accuracy and standard deviation computed over a moving window of the last five trials at each budget size, across different random seeds and hyperparameter samples drawn using Optuna. This experiment is based on the reviewer’s suggestion.
>
> As shown in **Table 1** and **Table 2**, TANDEM demonstrates stable accuracy across multiple hyperparameter trials, indicating that the method is not highly sensitive to hyperparameter tuning. This result holds for both datasets, where TANDEM is the best, and for those where it is not.
>
> ***Table 1**: mean Accuracy Over Trials on CP dataset (window of last 5 trials)*
> | Trial Budget | TabPFN        | XGBoost       | MLP           | TANDEM        |
> |--------------|---------------|---------------|---------------|---------------|
> | 5      | 0.592 ± 0.004 | 0.580 ± 0.004 | 0.575 ± 0.016 | 0.675 ± 0.011 |
> | 10     | 0.593 ± 0.002 | 0.574 ± 0.005 | 0.563 ± 0.012 | 0.678 ± 0.009 |
> | 20     | 0.597 ± 0.005 | 0.587 ± 0.005 | 0.564 ± 0.019 | 0.682 ± 0.005 |
> | 30     | 0.597 ± 0.004 | 0.580 ± 0.007 | 0.587 ± 0.007 | 0.682 ± 0.003 |
>
> ***Table 2**: mean Accuracy Over Trials on HI dataset (window of last 5 trials)*
>
> | Trial Budget | TabPFN        | XGBoost       | MLP           | TANDEM        |
> |--------------|---------------|---------------|---------------|---------------|
> | Trial 5      | 0.654 ± 0.006 | 0.628 ± 0.005 | 0.554 ± 0.020 | 0.640 ± 0.010 |
> | Trial 10     | 0.650 ± 0.002 | 0.633 ± 0.004 | 0.561 ± 0.017 | 0.642 ± 0.006 |
> | Trial 20     | 0.659 ± 0.004 | 0.633 ± 0.009 | 0.562 ± 0.006 | 0.644 ± 0.005 |
> | Trial 30     | 0.656 ± 0.005 | 0.632 ± 0.007 | 0.566 ± 0.014 | 0.644 ± 0.006 |
>
> **W2 + Q1: Hyperparameter transparency and computational efficiency**
>
> We clarify that all hyperparameter search ranges for TANDEM and the baselines are provided in the Appendix (Table E.1), ensuring transparency and reproducibility.
> To assess practical feasibility, we provide two tables below. **Table 3**  shows the mean pretraining time across all datasets for 50 epochs and 2000 samples per label on an L4 GPU.  TANDEM is faster than all self-supervised baselines evaluated, except for the self-supervised autoencoder (SS-AE), which happens because of the added OSDT encoder and gating net, which are both smaller than the SS-AE (and TANDEM’s) encoder-decoder network.
>
> ***Table 3**:  Pretraining Time per Model*
> | Model   | Mean Time (s) |
> |:--------|--------------:|
> | SS‑AE   | 38.12         |
> | TANDEM  | 43.47         |
> | TabNet  | 50.22         |
> | SCARF   | 63.05         |
> | SubTab  | 264.89        |
> | VIME    | 309.76        |
>
>
> **Table 4**  presents downstream training time per batch (128 samples) on an L4 GPU, where TANDEM behaves like a lightweight MLP.
> TANDEM's downstream training cost is minimal, reducing to a **simple MLP plus one-time usage of a gating network**, making it highly scalable and practical for real-world deployment, even on CPUs (in contrast to TabPFN). Its performance gains and representational power justifies the modest training overhead (just slightly more than the MLP).
>
> ***Table 4**:  Downstream Training Time per Batch*
>
> | Model             | Mean Time (s) |
> |-------------------|---------------:|
> | XGBoost           | 0.03           |
> | MLP               | 0.08           |
> | TANDEM | 0.08           |
> | DeepTFL           | 0.10           |
> | TabM              | 0.12           |
> | TabPFN            | 0.16           |
>
>
> **W3 + Q2 + Q4: Evaluation beyond classification and across labeling regimes**
>
> We thank the reviewer for pointing out the limited scope of the initial evaluation. Following this valuable feedback, we expanded our experiments in two significant directions:
> 1. **Extension to regression tasks**
>
> To test the generality of TANDEM beyond classification, we evaluated it on 13 regression datasets using 400 labeled samples per task, all from the benchmark from [1].
>
> The results, shown in **Table 5**, demonstrate that while less decisive than in the classification evaluation, TANDEM also achieves the best performance (lowest MSE and mean rank) compared to all baselines. We believe that results could be improved by using other losses for NN (as shown in [2]), and consider this an important area for future work.
>
> ***Table 5**:  mean MSE (MMSE)  and mean rank of various models across 13 regression datasets. Best results are bolded.*
>
> *Due to space constraints, detailed results for all evaluations in this response will be included in the appendix*.
> | Dataset       | CatBoost | LogReg |  MLP   | SCARF  | SubTab |  VIME  | TabM   | TabNet | XGBoost | SS AE  | TANDEM  |
> |---------------|----------|--------|--------|--------|--------|--------|--------|--------|---------|--------|---------|
> | **Mean MSE**  |  0.33  | 0.69 | 0.38 | 0.43 | 0.43 | 0.51 | 0.40 | 0.51 |  0.34 | 0.36 | **0.32** |
> | **Mean Rank** |   4.00   |  8.38  |  4.38  |  7.23  |  7.08  |  8.46  |  4.85  |  9.38  |   4.15  |  4.69  | **3.38**   |
>
> 2. **Varying labeled sample sizes**:
>
> We conducted additional experiments ranging from 50 to 1000 labeled samples per dataset on the classification task.
> As shown in **Table 6**, TANDEM consistently ranks at the top or remains competitive across all sample sizes. TANDEM achieves **state-of-the-art performance** with fewer than 800 labeled samples, and remains competitive even beyond that scale.
>
> ***Table 6**: mean classification accuracy for each model trained on different numbers of labeled samples. Bold values indicate the best performance for that sample size.*
>
> | Model    |     50 |    100 |    200 |    400 |    600 |    800 |   1000 |
> |:---------|-------:|-------:|-------:|-------:|-------:|-------:|-------:|
> | TabM     | 0.5572 | 0.6081 | 0.6281 | 0.6566 | 0.6847 | 0.6896 | 0.6920 |
> | SubTab   | 0.5787 | 0.6145 | 0.6348 | 0.6735 | 0.6788 | 0.6825 | 0.6846 |
> | SS-AE    | 0.6027 | 0.6298 | 0.6532 | 0.6799 | 0.6901 | 0.6735 | 0.6897 |
> | TabPFN   | 0.6150 | 0.6487 | 0.6734 | 0.6966 | 0.7183 | **0.7312** | **0.7374** |
> | TANDEM   | **0.6284** | **0.6591** | **0.6920** | **0.7151** | **0.7237** | 0.7284 | 0.7237 |
>
> These additions substantially strengthen the empirical foundation of the paper and highlight the significance of TANDEM’s contributions. The model consistently outperforms alternatives in both classification and regression tasks, across a wide range of supervision levels. While many public tabular datasets are large, real-world domains such as healthcare and finance often face limited labeled data. Our results demonstrate strong performance precisely in these challenging settings, underscoring the relevance and practical value of TANDEM. We thank the reviewer for raising this point, which prompted a more comprehensive and impactful evaluation.
>
> **W4: Feature correlation and reconstruction objectives**
>
> We agree that when features are weakly correlated, reconstruction-based pretraining may carry limited signal. However, in real-world tabular data, dependencies often exist between features. To demonstrate this, we computed the sum of absolute Pearson correlations between a feature and all other features (on numerical features only) within each dataset. In the table below, we present the mean of these values, which indicates substantial inter-feature dependencies in practice. We note that there may be additional nonlinear dependencies that have not been evaluated here.
>
> ***Table 7**: Mean and median of sum of absolute correlations per feature averaged across all datasets (numerical features only).*
> | Metric                         | Value (mean ± std)      |
> |-------------------------------|--------------------------|
> | Mean Sum Abs Corr             | 3.08 ± 1.61              |
> | Median Sum Abs Corr           | 2.03 ± 1.48              |
>
> **We hope this addresses all your concerns, and we would be happy to further clarify open issues.**
>
> [1] Grinsztajnet al. Why do tree-based models still outperform deep learning on typical tabular data?.
>
> [2] Stewart et al. Building Bridges between Regression, Clustering, and Classification.

---

> > ### Author Response · Authors · 2025-08-03
> > **Discussion period**
> >
> > We sincerely appreciate your time and valuable comments. Due to the limited discussion time, we would be grateful for any additional feedback or confirmation that our rebuttal addressed your concerns.

---

### Official Review · Reviewer_Fcpp · 2025-07-03

**Clarity:** 4
**Significance:** 3
**Originality:** 2
**Rating:** 5
**Confidence:** 5

**Summary:**

This paper proposes a new approach of learning from tabular data in low-label settings. Shortcomings of neural networks on tabular data are proposed to be mitigated by supplementing the modeling process with a tree-based model. This model intends to capture sharp variations commonplace in tabular data that neural networks have been shown to struggle on.

**Questions:**

1.  Are there any features that are known to be irrelevant/nuisance features in datasets from OpenML? Do we know which features are irrelevant in $\texttt{covertype}$ and $\texttt{PhishingWebsites}$?

2. Are irrelevant features and nuisance features the same? Is it possible that some features are a nuisance (to neural networks), but are not irrelevant? The way things are currently worded imply that OSDT is learning irrelevant features.

3. Is the gating network for OSDT selecting a higher proportion of categorical features compared to the gating network for the neural encoder?

**Ethical Concerns:**

["NO or VERY MINOR ethics concerns only"]

**Final Justification:**

The authors have sufficiently addressed my concerns in their responses and have communicated them clearly. I am increasing the rating to accept.

**Limitations:**

yes

**Quality:**

3

**Strengths And Weaknesses:**

Strengths:

•	This work bridges the research gap between modern deep learning techniques such as self-supervised learning, and applications that rely on tabular data.

•	Paper is well-written and organized.

•	Experiments to show that the gating networks lead to complimentary spectra is useful in making the case that the two models are not learning from the same set of features.

•	The dashed box to show the combined inductive biases an in Figure 1 is appreciated.

Weaknesses:

•	Lack of examples and experiments to test if existing feature selection and/or elimination methods fail on tabular data. If we can partition these feature sets, we can train a network and a tree separately, and then ensemble them.

•	It would be nice to have more explanation about how the complementary feature sets differ in terms of magnitude of dataset-specific overlap, and if models prefer certain types of features.

---

> ### Author Rebuttal · Authors · 2025-07-30
>
> Thank you for the thoughtful and well-engaged review. We appreciate your interest in the core motivation behind our work, especially your questions around how the gating networks behave and whether the two encoders prefer different types of features. We're also glad you found the paper clear and technically solid, and we appreciate your recognition of the strong empirical results. We're happy to address the insightful points you raised.
>
> **W1: Evaluating Feature Selection in Tabular Data**
>
> We thank the reviewer for raising the important question of whether existing feature selection or elimination methods are sufficient for inducing complementary representations in tabular learning. We address this in two parts: by evaluating **global feature selection with trees**, and by comparing TANDEM (our method) to **local masking-based self-supervised (SSL) approaches**.
>
> 1. **Global Feature Selection with Trees**:
>
> To explore the reviewer’s suggestion that partitioned feature sets could support effective ensemble, we evaluated whether applying explicit feature selection before training improves performance in a tree-based setup. Specifically, we trained a decision tree (DT) alongside a simple one-layer softmax model under two key settings: one using all input features, and one using only a subset selected by a SOTA NN-based model for feature selection (STG [1]).
> In both cases, ensembling improved performance over the tree alone. Notably, the variant with explicit FS yielded **further gains**, showing that even simple models benefit from selecting relevant features. While this setup achieved promising results competitive with strong classifiers like XGBoost and CatBoost, it still fell short of TANDEM.These findings support the reviewer’s intuition and highlight the value of **combining diverse inductive biases** through architectural integration.
>
> ***Table 1**: Accuracy of Decision Tree, FS-gated MLP, and their ensemble, along with Jaccard similarity between their top-10 selected features.*
>
> | Setting          | Tree Acc | Model Acc | Ensemble Acc | FeatSim |
> |:----------------:|---------:|----------:|-------------:|--------:|
> | with FS   |   0.630  |   0.644   |     **0.653**    |  **0.455**  |
> | without FS     |   0.630  |   0.622   |     0.641    |  0.405  |
>
>
> 2. **Evaluating Local Masking-Based Feature Selection Methods**:
>
> We aimed to evaluate how existing local feature selection methods in SSL compare to our approach. To this end, we considered two representative SSL methods that rely on masking: **VIME**, which applies noise to random subsets of features and trains the model to predict the corruption mask [2], and **SubTab**, which reconstructs inputs from masked subsets [3].
> To include a stronger baseline that incorporates learned feature selection, we also evaluated a self-supervised autoencoder with feature selection (SS-AE with FS using STG [1]), which learns to suppress irrelevant features during pretraining.
>
> As shown in **Table 2**, both VIME and SubTab underperform compared to TANDEM, demonstrating the limitations of random or task-agnostic masking used for local feature selection in SSL. In contrast, SS-AE with FS performs significantly better than the other local feature selection methods, highlighting the value of learned, sample-specific gating in self-supervised settings.
>
> Still, TANDEM consistently outperforms all alternatives, suggesting that its **combination of gating with diverse encoder types** is key to learning strong, complementary representations.
>
> ***Table 2**: Mean accuracy and mean rank across datasets for VIME, SubTab, SS-AE with STG, and TANDEM.
> All methods were evaluated on the same datasets and under the same evaluation protocol as used in the paper, with identical pretraining data.*
> *Due to space constraints, detailed results for all evaluations in this response will be included in the appendix*.
>
> | Model        | Mean Accuracy | Mean Rank |
> |--------------|----------------|------------|
> | TANDEM       | **0.7157**     | **1.40**   |
> | SS-AE with FS  | 0.6941         | 2.20       |
> | SubTab       | 0.6735         | 3.10       |
> | VIME         | 0.6489         | 3.70       |
>
> **W2 + Q3: Complementary gating behavior**:
>
>  To further analyze how the neural and tree components differ in their learned feature preferences, we computed several metrics across four datasets with a moderate proportion of categorical features (**Table 3** , further details on the datasets categorization are shown in the response to reviewer **Meqn**). This analysis focuses specifically on how the gating network of the decision tree and the gating network of the neural network differ in their feature selection behavior over categorical features. We report three interpretable metrics. **Gate Overlap** measures the proportion of categorical features that are simultaneously activated by both gating networks. **Gate Correlation** captures the alignment between the gate strength values assigned by the two networks using Pearson correlation. **Gate Variance Ratio (Tree / NN)** reflects how sharply each gating network modulates feature importance by comparing the variance in their gate strengths. We observe moderate agreement between the gating networks in terms of overlap and correlation. Furthermore, the tree’s gating network consistently shows **higher variance**, indicating it makes more decisive distinctions between categorical features. In contrast, the neural gating network applies smoother, more conservative gating. These differences support our core design goal in TANDEM: to encourage the two components to attend to **complementary aspects** of the input.
>
>  ***Table 3**: Comparison of gating behavior between the neural and tree encoders, evaluated only on the categorical features of the datasets.*
>
>
> | Dataset | Gate Overlap | Gate Correlation | Gate Variance Ratio (Tree / NN) |
> |---------|--------------|------------------|----------------------------------|
> | RS      | 0.78         | 0.88             | 1.52                             |
> | ALB     | 0.59         | 0.66             | 1.81                             |
> | EY      | 0.33         | 0.34             | 1.61                             |
> | CC      | 0.65         | 0.82             | 1.76                             |
>
>
> **Q1: Feature relevance in OpenML datasets**
>
> We did not find official documentation labeling irrelevant or nuisance features in the OpenML datasets. Globally, most features appear task-relevant; however, we believe some features act as local nuisances, i.e., unhelpful for specific samples even if not globally irrelevant. This is supported by the behavior of our gating mechanism, which selectively downweights such features during inference.
> Similar observations have been made in prior work, where both global and local nuisance features were identified and mitigated([1], [4]).
>
> Further analysis of feature quality is crucial to ensure these benchmarks remain both informative and realistic.
>
> **Q2: OSDT robustness to nuisance features**
>
>  We refer to nuisance features as variables that may be irrelevant for the prediction task. For example, in genetics, when trying to predict a medical condition, often only a small subset of variables carries information, while the rest are nuisance features. In our current notation, we treat nuisance and irrelevant features interchangeably.
>
> To clarify our claim that tree-based methods are more resilient to nuisance features, we refer to prior work [5], which shows that decision trees are inherently less sensitive to uninformative features due to their greedy, axis-aligned splits and implicit regularization.
> To further illustrate this property empirically, we injected 100 purely irrelevant Gaussian features into the CO dataset. As shown in Table 4, XGBoost’s performance remained stable, whereas the MLP experienced a significant drop in accuracy. This validates the notion that nuisance features can degrade the performance of NNs and underscores the robustness of tree-based approaches, like OSDT, in attenuating the influence of nuisance features.
>
> ***Table 4**: Impact of adding 100 Gaussian nuisance features to the CO dataset.*
>
> | Dataset | Model   | Accuracy (original)     | Accuracy (with noise)     |
> |---------|---------|-------------------------|----------------------------|
> | CO      | XGBoost | 0.5326 ± 0.016          | 0.5317 ± 0.018            |
> | CO      | MLP     | 0.4963 ± 0.014          | 0.4437 ± 0.029            |
>
> **We appreciate the feedback and would be happy to further clarify any open issues.**
>
> [1] Yamada et al. Feature Selection using Stochastic Gates.
>
> [2] Yoon et al. VIME: Extending the success of self- and semi-supervised learning to tabular domain.
>
> [3] Ucar et al. SubTab: Subsetting features of tabular data for self-supervised representation learning.
>
> [4] Yoon et al. INVASE: Instance-wise Variable Selection using Neural Networks
>
> [5] Grinsztajnet al. Why do tree-based models still outperform deep learning on typical tabular data?

---

> > ### Author Response · Authors · 2025-08-03
> > **Discussion period**
> >
> > We sincerely appreciate your time and valuable comments. Due to the limited discussion time, we would be grateful for any additional feedback or confirmation that our rebuttal addressed your concerns.

---

> > ### Comment · Reviewer_Fcpp · 2025-08-06
> >
> > Thanks for looking into these issues in such detail. I understand that space is limited but I think that these information will help readers to understand the motivations behind the research problem.

---

> > > ### Author Response · Authors · 2025-08-07
> > > **Thank you for your engagement and insightful feedback**
> > >
> > > We sincerely thank you for your encouragement and constructive feedback. We fully agree that the additional experiments and results can help clarify the motivations behind our approach, and we will include them in the main text and  appendix of the final version. We appreciate your thoughtful engagement throughout the review process.

---

### Official Review · Reviewer_Meqn · 2025-07-06

**Clarity:** 3
**Significance:** 2
**Originality:** 3
**Rating:** 4
**Confidence:** 4

**Summary:**

This paper introduces TANDEM (Tree-And-Neural Dual Encoder Model), which combines neural networks with oblivious soft decision trees in a hybrid autoencoder for tabular classification under limited labels. The key idea is using two complementary encoders - a standard neural network and a tree-based encoder - each with sample-specific gating networks that act as learnable feature selectors. The encoders share a decoder and train jointly via reconstruction, alignment, and latent similarity losses. At test time, only the neural encoder is used. The authors test this on 19 OpenML datasets using 400 labeled examples each, reporting improvements over various supervised baselines.

**Questions:**

1. Why exclude SCARF, VIME, and SubTab from comparisons? These are the most relevant recent tabular SSL methods and their absence is puzzling given they're discussed in related work.

2. Can you explain the TabNet baseline results? The 0.54 mean accuracy seems much lower than expected for this method. What hyperparameters were used and how was it implemented?

3. What about statistical significance? With 100 runs, you should be able to provide p-values for the reported improvements. Are these differences statistically meaningful?

4. How does this scale? What happens with larger datasets (>10K samples) or different label ratios? Is the approach computationally practical for realistic problem sizes?

5. Which datasets benefit most? Is there any pattern in terms of dataset characteristics (number of features, categorical vs numerical, etc.) that determine when TANDEM helps most?

**Ethical Concerns:**

["NO or VERY MINOR ethics concerns only"]

**Final Justification:**

I thank the authors for the detailed responses! After evaluating the responses to all of the reviews, I have decided to increase the rating due to the added value and insight through the additional experiments.

**Limitations:**

The authors mention small sample sizes and low-label focus, but several other limitations deserve attention:

Evaluation scope: Results from 19 carefully filtered datasets with 400 examples each may not generalize to the broader space of tabular problems.

Baseline coverage: Missing comparisons to the most relevant existing methods limits our understanding of true performance gains.

Hyperparameter sensitivity: The method introduces multiple hyperparameters (loss weights, gating noise, tree parameters) but there's limited analysis of sensitivity to these choices.

Real-world applicability: The controlled experimental setting may not reflect challenges in practical deployment.

**Quality:**

2

**Strengths And Weaknesses:**

Strengths

Originality: Genuinely novel architectural contribution.
The hybrid neural-tree autoencoder for tabular SSL appears unprecedented in the literature. While NODE uses oblivious decision trees and various methods combine neural/tree approaches, no prior work integrates them within a self-supervised autoencoder framework. The "model-based augmentation" concept represents a paradigm shift from data corruption approaches (SCARF, VIME, SubTab) to architectural diversity, which could influence future research directions.

Quality: Technically sound with well-motivated design.
The three-component loss function is well-designed: reconstruction loss preserves semantic information, alignment loss (L2 distance between reconstructions) prevents encoder divergence, and cosine similarity in latent space maintains consistency despite architectural differences. The stochastic gating mechanism using clipped Gaussian noise (σ=0.5) provides differentiable feature selection. The spectral analysis provides concrete evidence that neural gating acts as a low-pass filter while tree-based gating preserves high-frequency content, validating the complementary inductive biases claim.

Clarity: Well-structured presentation with clear motivation.
The paper effectively motivates the problem (neural networks' spectral bias on tabular data), clearly explains the hybrid architecture, and provides intuitive explanations for design choices. Mathematical notation is consistent, figures effectively illustrate the approach, and the spectral analysis adds valuable theoretical insight. The writing flows logically from problem identification through method description to experimental validation.

Significance: Addresses important limitations in tabular SSL.
The work tackles a real problem - neural networks often underperform trees on tabular data, and existing tabular SSL methods struggle with effective augmentation strategies. The approach could help bridge the gap between tree-based and neural methods for tabular learning, potentially improving performance in low-label regimes where labeled data is expensive.


Weaknesses

Quality: Experimental validation is severely limited.
The evaluation scope is problematically narrow - only 19 datasets with artificially constrained settings (≥2,500 samples per class, exactly 400 labeled examples) cannot support general effectiveness claims. Missing comparisons to key tabular SSL methods (SCARF, VIME, SubTab) mentioned in related work make it impossible to assess true advancement. Baseline results are concerning: TabNet achieving 0.54 mean accuracy while TANDEM gets 0.71 seems implausible without implementation details. No statistical significance testing, despite 100 repetitions, undermines confidence in reported improvements.

Clarity: Key experimental details are underexplained.
While the method description is clear, critical experimental aspects lack transparency. Baseline implementation details are missing, making suspicious results (TabNet's poor performance) hard to evaluate. The choice of hierarchical gating for trees versus single gating for neural encoders is not well justified. Computational cost analysis is absent, leaving scalability questions unanswered.

Significance: Limited demonstrated impact.
The narrow experimental scope (400 labeled examples on filtered datasets) severely limits practical relevance. Many real-world tabular problems involve thousands of samples, not hundreds. Without broader evaluation or comparison to recent SSL methods, it's unclear whether this approach offers meaningful advantages over existing techniques. The improvements, while consistent in the constrained setting, may not generalize to realistic scenarios.

Originality: Individual components largely exist.
While the specific combination is novel, core components are incremental extensions of existing work: oblivious decision trees (NODE), stochastic gating (similar to TabNet's attention), autoencoder SSL (standard), and multi-component losses (common in contrastive learning). The novelty lies primarily in integration rather than fundamental algorithmic innovation. The model-based augmentation concept, while interesting, needs broader validation to demonstrate its effectiveness beyond this specific architecture.

---

> ### Author Rebuttal · Authors · 2025-07-30
>
> We thank the reviewer for their thoughtful assessment and constructive feedback. We are particularly grateful for your recognition of TANDEM’s “genuinely novel architectural contribution” and its “paradigm shift” in self-supervision. Your appreciation of our well-motivated design, including the alignment and latent similarity losses, as well as the spectral analysis that validates the complementary inductive biases, is deeply encouraging. We also value your acknowledgment of the vital challenge our work addresses by bridging the gap between neural and tree-based models in tabular learning.
>
> We found your comments and questions extremely helpful, and we address each of them below, providing additional experiments, clarifications, and analyses.
>
>
> **W1 + W3 + Q1: Experimental validation and scope**
>
>
> We thank the reviewer and want to clarify that our method’s contributions extend beyond specific dataset sizes or label counts. Nonetheless, we have significantly expanded the experiments in three key directions:
>
> 1. **Recent SSL baselines**:
> We added SCARF, VIME, and SubTab to our benchmark, evaluating them on the same 19 classification datasets with 400 labeled samples per class.
> As shown in **Table 1**, TANDEM outperforms all three methods in both mean accuracy and mean rank across datasets.
> This confirms that our hybrid approach with dual inductive biases provides stronger representations than existing SSL alternatives in low-label regimes.
>
> *Table 1: mean accuracy and mean rank per over all datasets*
>
> *Due to space constraints, detailed results for all evaluations in this response will be included in the appendix*.
>
> | Dataset           | SCARF      | VIME       | SubTab     | TANDEM     |
> | ----------------- | ---------- | ---------- | ---------- | ---------- |
> | **Mean Accuracy** | 0.6696     | 0.6489     | 0.6735     | **0.7157** |
> | **Mean Rank**     | 3.00       | 3.11       | 2.58       | **1.32**   |
>
> 2. **Extension to regression tasks**:
> To test the generality of TANDEM beyond classification, we evaluated it on 13 regression datasets using 400 samples per task, all from the benchmark from [1].
> The results, shown in **Table 2**, demonstrate that TANDEM also achieves the best average performance (lowest mean MSE and mean rank) compared to all baselines. We believe that results could be improved by using other losses for NN (as shown in [2]), and consider this an important area for future work.
>
> ***Table 2**:  mean MSE (MMSE)  and mean rank of various models across 13 regression datasets. Best results are bolded.*
>
> | Dataset       | CatBoost | LogReg |  MLP   | SCARF  | SubTab |  VIME  | TabM   | TabNet | XGBoost | SS AE  | TANDEM  |
> |---------------|----------|--------|--------|--------|--------|--------|--------|--------|---------|--------|---------|
> | **Mean MSE**  |  0.33  | 0.69 | 0.38 | 0.43 | 0.43 | 0.51 | 0.40 | 0.51 |  0.34 | 0.36 | **0.32** |
> | **Mean Rank** |   4.00   |  8.38  |  4.38  |  7.23  |  7.08  |  8.46  |  4.85  |  9.38  |   4.15  |  4.69  | **3.38**   |
>
>
> 3. **Varying the number of labels**:
> We conducted additional experiments ranging from **50 to 1000** labeled samples per dataset on the classification task.
> As shown in **Table 3**, TANDEM consistently ranks at the top or remains competitive across all sample sizes.
>
> ***Table 3**: mean classification accuracy for each model trained on different numbers of labeled samples. Bold values indicate the best performance for that sample size (we report only the top 5 models here).*
>
> | Model    |     50 |    100 |    200 |    400 |    600 |    800 |   1000 |
> |:---------|-------:|-------:|-------:|-------:|-------:|-------:|-------:|
> | TabM     | 0.5572 | 0.6081 | 0.6281 | 0.6566 | 0.6847 | 0.6896 | 0.6920 |
> | SubTab   | 0.5787 | 0.6145 | 0.6348 | 0.6735 | 0.6788 | 0.6825 | 0.6846 |
> | SS-AE    | 0.6027 | 0.6298 | 0.6532 | 0.6799 | 0.6901 | 0.6735 | 0.6897 |
> | TabPFN   | 0.6150 | 0.6487 | 0.6734 | 0.6966 | 0.7183 | **0.7312** | **0.7374** |
> | TANDEM   | **0.6284** | **0.6591** | **0.6920** | **0.7151** | **0.7237** | 0.7284 | 0.7237 |
>
>
> These additions substantially strengthen the empirical foundation of the paper and highlight the significance of TANDEM’s contributions. The model consistently outperforms alternatives in both classification and regression tasks, across a wide range of supervision levels. While many public tabular datasets are large, real-world domains such as healthcare and finance often face a shortage of labeled data. Our results demonstrate strong performance precisely in these challenging settings, underscoring TANDEM’s relevance and practical value (see e.g., [3],[4]). We thank the reviewer for raising this point, which prompted a more comprehensive and impactful evaluation.
>
> **W1 + Q3: Statistical significance comparison**
>
> ***Table 4**: how many datasets (out of 19; 18 for TabPFN) TANDEM outperforms each baseline. Parentheses indicate **statistically significant** wins (p < 0.05, 100 trials).*
>
> |           | MLP     | XGBoost | TabM    | MLogReg | DeepTLF | CatBoost | SS-AE   | TabPFN   | VIME     | SCARF    | SubTab   |
> |-----------|---------|---------|---------|---------|---------|----------|---------|----------|----------|----------|----------|
> | TANDEM    | 18 (17) | 18 (17) | 18 (17) | 19 (18) | 19 (19) | 19 (17)  | 18 (18) | 13 (14)  | 16 (17)  | 17 (19)  | 16 (16)  |
>
>
> **W2 + Q2: TabNet baseline results**
>
> We understand the reviewer’s concern regarding the TabNet results. We used the **pytorch-tabnet** package and tuned its full pipeline (pretraining + classification/regression) via Optuna. In addition, we reproduced the official TabNet implementation from GitHub, including its embedding tuning. Both setups were tuned independently; the best achieved ~0.57 mean accuracy, which is below the performance of TANDEM.
> Prior work [5] also reported suboptimal TabNet performance on tabular data, including datasets from our benchmark, likely due to its sensitivity to tuning and limited handling of categorical features. We thank the reviewer for raising this concern.
> All implementation and optimization details appear in the appendix.
>
>
> **W2: Justification for hierarchical vs. shared Gating**
>
> In the **tree encoder**, each layer operates directly over the **original input feature space**. Because of this, **gating must be applied at every depth**, to determine which features are relevant for that layer’s decisions. Moreover, to preserve the **oblivious** structure where layers operate independently, we use a **separate** gating network for each layer. Each of these gating networks is tailored specifically to its corresponding layer, making the overall mechanism inherently hierarchical.
>
> In contrast, the **neural encoder** applies gating once at the input, since subsequent layers operate **sequentially on the resulting feature embeddings**.
>
> **Q4: Computational cost and scalability**
>
> During inference, TANDEM is applied as a fully connected MLP, making it highly efficient and lightweight, suitable even for home CPU environments. Pretraining is under 2× the cost of the self-supervised Autoencoder, with the primary overhead from the tree encoder, which scales modestly with depth. Despite this, TANDEM remains significantly faster than SubTab, VIME, TabNet, and SCARF, both in training and inference.
>
> **Q5: Categorical feature ratio breakdown**
>
> To better understand where TANDEM excels, we grouped the datasets by their categorical feature ratio.
>
> **High (≥ 0.70):** PW (1.00), BM (0.92), CO (0.85), AD (0.87)
> **Medium (0.30–0.69):** RS (0.69), ALB (0.39), EY (0.33), CC (0.33)
> **Low (< 0.30):** OG (0.10), CP (0.27), VO (0.18), EL (0.12), HI, HE, JA, NU, CA (0.00)
>
> This characterization highlights that TANDEM performs particularly well on datasets with a **moderate share of categorical features** and also shows strong results on entirely numeric datasets with **low-cardinality continuous features**, which often behave like categorical inputs.
> Studying this effect further is an interesting direction for future work.
>
> **L4: Sensitivity analysis of TANDEM hyperparameters**
>
> We conducted a sensitivity analysis over five datasets, examining various ranges of loss weights, gating hidden dimension, tree depth, and number of trees. TANDEM showed **stable performance** across a wide range of values. Optimization trials using Optuna on two datasets (one where TANDEM excels and one where it does not) demonstrated **stable performance**.
> Additional results will be included in the appendix. The alignment and latent representation similarity loss weights sensitivity analysis is discussed in the response to reviewer **uEwZ**, and the optimization behavior is addressed in response to reviewer **km3R**. (Omitted here due to space constraints).
>
> **We hope this addresses all your concerns, and we would be happy to further clarify open issues.**
>
> [1] Grinsztajn  et al. Why do tree-based models still outperform deep learning on typical tabular data?
>
> [2] Stewar et al. Building Bridges between Regression, Clustering, and Classification.
>
> [3] Arazi at al. TabSTAR: A Foundation Tabular Model With Semantically Target-Aware Representations.
>
> [4] Hollmann et al. TabPFN: A Transformer That Solves Small Tabular Classification Problems in a Second.
>
> [5] Gorishniy et al Revisiting deep 318 learning models for tabular data.

---

> > ### Author Response · Authors · 2025-08-03
> > **Discussion period**
> >
> > We sincerely appreciate your time and valuable comments. Due to the limited discussion time, we would be grateful for any additional feedback or confirmation that our rebuttal addressed your concerns.

---

### Note · Authors · 2025-08-13

Dear Area Chairs and Reviewers,

We thank you for a constructive review process. The discussion period prompted improvements in clarity, empirical scope, and analysis.

### **Novelty and core contribution**
We clarified that **TANDEM is not an ensemble**: the tree encoder is used only during training to inject complementary inductive bias into the neural encoder via shared objectives and alignment losses. At inference, only the neural encoder remains. This defines a novel **model-based augmentation** paradigm, where a structured model guides neural representation learning in SSL. Coupled with **per-encoder stochastic gating**, TANDEM dynamically modulates encoder contributions during training, enabling richer, more complementary feature extraction.

### **Expanded empirical validation**
- **Additional baselines:** Added SCARF, SubTab, and VIME (as requested in the review) to broaden the SSL comparison. **TANDEM outperforms all SSL baselines across 19 datasets.**
- **Regression tasks:** On 13 regression datasets from the same benchmark, **TANDEM achieves the best mean rank** and lowest mean MSE, with significant gains over most baselines.
- **Varying labels:** From 50–1000 labels, TANDEM consistently ranks at or near the top, demonstrating robustness beyond the 400-label setting.
- **Statistical significance:** Classification and regression gains are statistically significant over all baselines (p < 0.05).

### **Analysis and interpretability**
- **Gating Mechanism:** In addition to spectral analysis, quantitative metrics show complementary gating patterns. Trees make higher-variance, more decisive selections, while neural gating is smoother, aligning with our inductive bias.
- **Ablations:** Removing gating or alignment losses consistently reduces performance.

### **Stability**
TANDEM maintains strong performance after 15–20 iterations and is robust across wide range of consistency loss weights, showing **stability** and **low hyperparameter sensitivity**.

### **Efficiency and scalability**
TANDEM trains faster than most SSL baselines and matches MLP inference cost, enabling CPU use.

These clarifications, extended experiments, and statistical tests strengthen the case for TANDEM’s **novelty, practical relevance, and generalization**. We believe the principle of model-based augmentation can benefit a wide range of architectures for tabular SSL, and we hope these advances are reflected in your final assessments.

---

### Decision · Program_Chairs · 2025-09-17

**Decision:**

Accept (poster)

**Comment:**

The paper provides a solution for tabular data tasks in low-label settings. It is mentioned to be well written, and provide a practical solution to an important problem (Meqn, “The work tackles a real problem - neural networks often underperform trees on tabular data, and existing tabular SSL methods struggle with effective augmentation strategies.” uEwZ ”The method does not rely on complex pretraining or specialized architectures, making it easy to implement and extend.”).

In addition to the bottom-line results showing a competitive performance compared with SOTA techniques, the reviews appreciated the studies motivating the different design choices. km3R “The complementarity of the inductive biases of the two components is thoroughly demonstrated in the experiments,” , Meqn “The spectral analysis provides concrete evidence that neural gating acts as a low-pass filter while tree-based gating preserves high-frequency content, validating the complementary inductive biases claim.”, Fcpp “Experiments to show that the gating networks lead to complimentary spectra is useful in making the case that the two models are not learning from the same set of features.”.

The paper initially had a borderline rating by the reviewers mostly due to incomplete experiments, requiring an extension of the study to regression tasks in addition to classification tasks, and various ablation studies. The authors provided these experiments in the rebuttal stage and the reviewers Fcpp and km3R raised their score, mentioning that their concerns were fully mitigated and in particular these experiments provide sufficient justification to the design choices and proof of the competitiveness of the method compared to existing art.

A remaining issue without consensus is the novelty and insights provided by the paper. Despite the rebuttal, the reviewers had mixed opinions about the subject, leading to different overall scores. This is a more subjective issue, and given that more than one reviewer did find the work to have interesting insights and novel components, I believe that the paper will be of value to participants of NeurIPS.